# Switching between H- and J-type electronic coupling in single conjugated polymer aggregates

Theresa Eder[1], Thomas Stangl[1], Max Gmelch[1], Klaas Remmerssen[2], Dirk Laux[2], Sigurd Höger[2], John M. Lupton[1] & Jan Vogelsang [iD] [1]

The aggregation of conjugated polymers and electronic coupling of chromophores play a central role in the fundamental understanding of light and charge generation processes. Here we report that the predominant coupling in isolated aggregates of conjugated polymers can be switched reversibly between H-type and J-type coupling by partially swelling and drying the aggregates. Aggregation is identified by shifts in photoluminescence energy, changes in vibronic peak ratio, and photoluminescence lifetime. This experiment unravels the internal electronic structure of the aggregate and highlights the importance of the drying process in the final spectroscopic properties. The electronic coupling after drying is tuned between H-type and J-type by changing the side chains of the conjugated polymer, but can also be entirely suppressed. The types of electronic coupling correlate with chain morphology, which is quantified by excitation polarization spectroscopy and the efficiency of interchromophoric energy transfer that is revealed by the degree of single-photon emission.

[1] Institut für Experimentelle und Angewandte Physik, Universität Regensburg, Universitätsstraße 31, 93053 Regensburg, Germany. [2] Kekulé-Institut für Organische Chemie und Biochemie, Universität Bonn, Gerhard-Domagk-Straße 1, 53121 Bonn, Germany. Correspondence and requests for materials should be addressed to J.V. (email: jan.vogelsang@physik.uni-regensburg.de)

"**A** primary mass is a number of individuals who have put one and the same object in place of their ego ideal and consequently identify with each other." reads the famous definition of Sigmund Freud regarding a collective of humans[1]. Loosely, this description also holds true for less complicated systems such as aggregates of single molecules, which can form an ordered mesoscopic object where all incorporated molecules behave as one thanks to coherent electronic coupling between molecular electronic excitations[2–5]. Such coherent coupling phenomena play a significant role in biological photosynthetic systems by supporting an efficient transport of energy from light-harvesting antennas to photosynthetic reaction centers[6–10]. For this reason, coherence effects in photosynthetic systems have inspired their usage in photovoltaic concepts which has driven the need for a fundamental understanding of coherent coupling in soft organic assemblies, such as conjugated polymers (CPs)[11]. The theoretical framework to describe the coherent coupling mechanisms in CP was put forward by Spano et al. and is built upon a combination of J-type and H-type couplings: unconventional J-type coupling between covalently coupled repeat units within a polymer chain rather than conventional J-aggregation in van-der-Waals bound aggregates, and conventional H-type coupling between cofacial chromophoric units on neighboring chains which abides by Kasha's exciton theory[3,12–15]. Figure 1a depicts three different cases: H-type, J-type, and the absence of coherent coupling in CP aggregates, along with their expected relationships with chain morphology. J-type coupling occurs predominantly along the same CP chain, because of the covalent interactions between the head-to-tail arranged transition dipole moments (TDMs) of the repeat units[14,16]. If more repeat units are coupled together or if the coupling strength increases between repeat units, e.g., by planarization of the chain, which leads to improved π-conjugation between repeat units, a stronger J-type photoluminescence (PL) will be observed. In terms of the spectroscopic observables, J-aggregation in van-der-Waals bound structures and improved conjugation in covalently bound polymer repeat units are effectively equivalent[17,18]. Generally, most conjugated polymers can, to a first approximation, be thought of as direct-gap semiconductors. In such systems the effective exciton band curvature around $k = 0$ (i.e. at zero momentum) is positive, which is the requirement for J-type photophysics[19]. The transition from the lowest excited state to the ground state of such a J-type coupled system is highly dipole-allowed and leads to the following associated spectroscopic changes of the photoluminescence (PL) compared to an uncoupled system: increased radiative rate and therefore a reduced PL lifetime, spectral red-shift, and narrowing of the 0–0 vibronic transition, and increased 0–0 to 0–1 vibronic peak ratio[14,20,21]. One particularly prominent example for this J-type behavior in CPs is the effectively defect-free trans-polydiacetylene isolated chain. This material shows an effective ratio of 0–0 to 0–1 PL peaks of almost one hundred[18,22], a strong indication of excitonic delocalization of order 100 nm, due to J-type coupling[23]. H-type coupling, on the other hand, occurs mainly between chromophoric units on different CP chains, because a side-by-side arrangement is required. Here, the transition from the lowest excited state to the ground state is, in principle, dipole forbidden, hence the opposite changes in PL are expected in comparison to J-type coupling[3,24–27]. For an H-type aggregate, emission occurs through a weak excimer transition, the difference to a gas-phase excimer being that the dimer structure still exists in the ground state[27–29]. Intuitively, one expects different nanoscale morphologies to be responsible for both coupling mechanisms and the following hypothesis can be formulated: for substantial J-type coupling to occur, intrachain ordering must be present to align the TDMs of neighboring repeat units in a head-to-tail fashion, as depicted in the left and

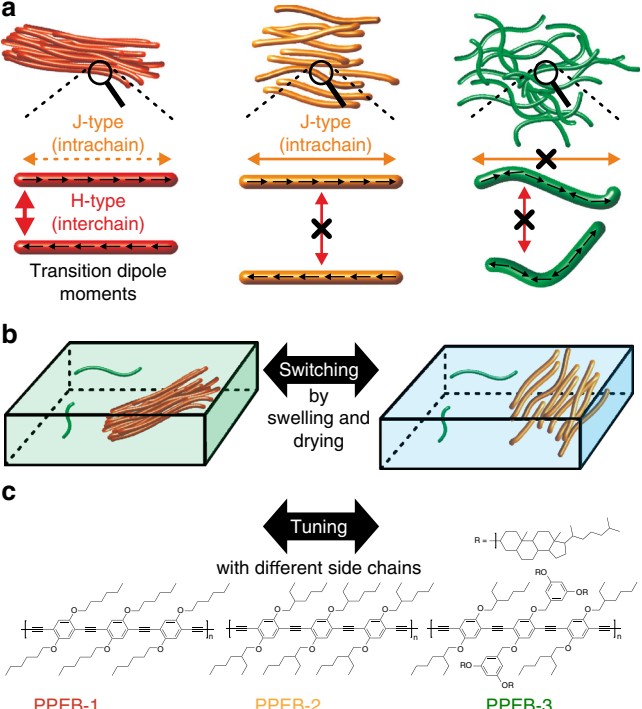

**Fig. 1** Switching and tuning between different types of electronic coupling. **a** Mechanisms of H-type and J-type coupling and the suppression of interactions and their expected relationship with aggregate morphology. **b** Switching between H-type and J-type coupling by drying (green box) and swelling (blue box) the aggregate embedded inside a polymer matrix by solvent vapor annealing. **c** Structures of samples used to tune between H-type (PPEB-1), J-type (PPEB-2), and suppressed coupling (PPEB-3) by changing the side chains

middle panel of Fig. 1a. Additionally, a high degree of interchain ordering with small spatial separations between neighboring chains is required for H-type coupling to occur, as depicted in the left panel of Fig. 1a. Therefore, switching between these two different forms of electronic coupling becomes feasible if the nanoscale morphology is controlled by either external or internal handles, for example by changing the surrounding environment or the molecular structure.

Unfortunately, testing this hypothesis in bulk measurements is almost always very difficult, because of the extraordinary morphological heterogeneity of CPs. Single-molecule spectroscopy (SMS) is capable of unraveling this heterogeneity and testing the relationship between intrachain ordering and J-type coupling[30–32]. However, interchain interactions are neglected in SMS. We recently approached this predicament by growing isolated aggregates of CPs by solvent vapor annealing (SVA) to overcome the inherent averaging over differently aggregated regions in bulk measurements and simultaneously include interchain interactions[33–35]. In a previous publication we demonstrated the evolution of electronic coupling by deterministically growing aggregates out of single CP chains by SVA[35]. As a model system we used poly(para-phenylene-ethynylene-butadiynylene) (PPEB) bearing hexyloxy side chains with a number-average molecular weight of $M_n = 40$ kDa and a polydispersity index (PDI) of 1.46. The structure is shown in Fig. 1c and denoted here as PPEB-1. The coupling is identified by a 10-fold increase in PL lifetime, a strong decrease of the 0–0 to 0–1 peak ratio and a corresponding spectral red-shift. This observation leads to the conclusion that such aggregates indeed exhibit pronounced H-type coupling, giving rise to enhanced energy

transfer, which in turn results in deterministic single-photon emission[35].

Here, we test the above hypothesis by switching between H-type and J-type coupling in aggregates grown by SVA. The basic approach is depicted in Fig. 1b. The aggregates are partially swollen by fine tuning the SVA process with a solvent mixture to prevent them disintegrating into single CP chains. Swelling cancels the interchain H-type coupling. By following the shift in PL, the change of vibronic peak ratios and the PL lifetime of isolated aggregates in situ, we demonstrate that the dominant electronic coupling can be switched reversibly between H-type and J-type.

This observation confirms the predicted simultaneous presence of H-type and J-type electronic coupling in CPs, and demonstrates that H-type coupling can completely mask the increased intra-chain J-type coupling arising from improved chain ordering in the aggregates. Finally, we demonstrate how small changes in the molecular structure of side chains of the CPs, as shown in Fig. 1c, can be exploited to tune between H-type (PPEB-1), J-type (PPEB-2), and suppressed coupling (PPEB-3) in the dried state. The different types of electronic coupling are correlated with chain morphology in the aggregate, which is determined by excitation polarization spectroscopy, and the energy transfer properties within the aggregate, which are assessed by the degree of single-photon emission.

## Results

**Switching between H-type and J-type coupling.** CPs are diluted to ten times above single-molecule concentration (i.e., $\sim 10^{-9}$ M) and embedded in a non-fluorescent poly(methyl-methacrylate) (PMMA) matrix of 200–250 nm thickness on top of a glass substrate by spin coating. The single CP chains are uniformly distributed inside the matrix. Subsequently, the film is annealed with a mixture of chloroform and acetone vapor by purging nitrogen through a chloroform and acetone reservoir, respectively, and transporting the chloroform/acetone saturated nitrogen vapor onto the thin polymer film. This leads to swelling of the film and diffusion of the single CP chains[36], enabling them to aggregate with each other to form mesoscopically sized objects consisting of multiple CP chains[33]. The spectroscopic properties of single CP chains and aggregates consisting of PPEB-1 are known and both objects can be easily distinguished by their different PL emission energies[35]. Whereas the PL emission from single chains is dominated by a 0–0 peak position just above 2.5 eV, the aggregates emit mostly below 2.5 eV with a strong reduction of the 0–0 peak due to efficient H-type coupling[35]. By counting the number of spots before and after SVA, we assess the average number of CP chains in an aggregate[35]. The following experiments are performed on aggregates of 15 single chains on average.

After the SVA aggregation process is completed the films are imaged with a wide-field fluorescence microscope setup equipped with a dual-view EMCCD-camera to identify the positions and emission colors of the aggregates and left-over single CP chains. The sample is excited by a fiber-coupled diode laser with an

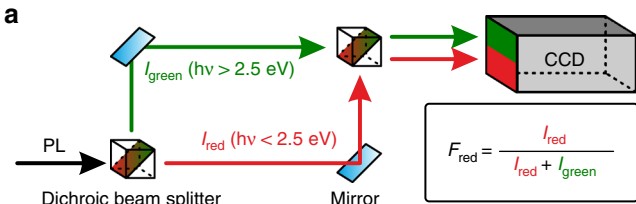

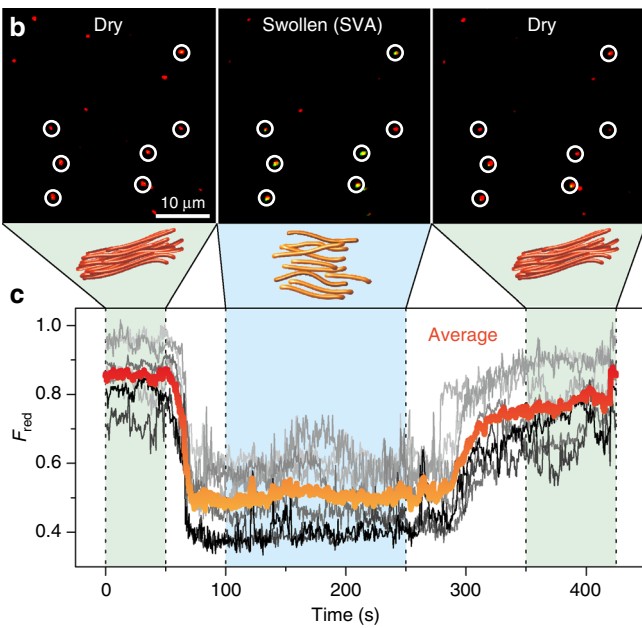

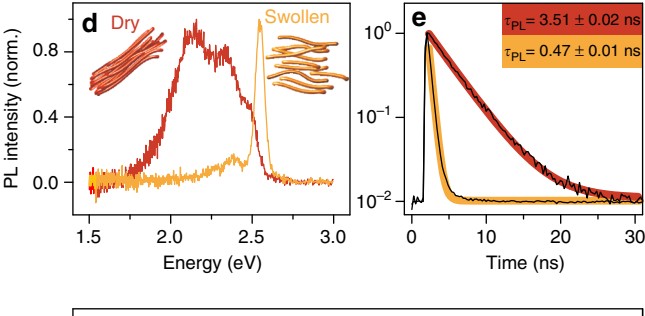

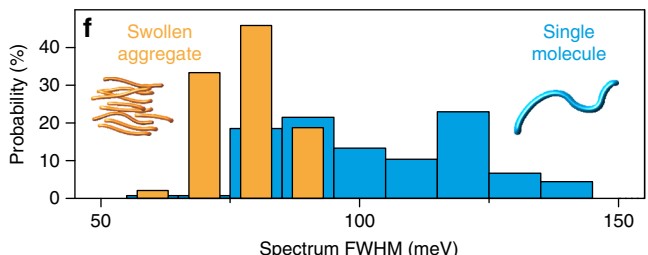

**Fig. 2** Switching between H-type and J-type coupling using solvent vapor annealing. **a** Schematic illustrating the splitting of single-spot PL with a dual-view unit into two detection channels, for photons with $h\nu > 2.5$ eV (denoted $I_{green}$) and $h\nu < 2.5$ eV (denoted $I_{red}$), and imaging onto different areas on the same CCD camera. The fraction of red emission for each single spot is defined as $F_{red}$. **b** Accumulated wide-field dual-view fluorescence microscopy images of PPEB-1 aggregates of the same area from a 425 s movie during which the film conditions were changed. The first frame is accumulated during the first 50 s, the second frame at time 100–250 s and the last frame at 350–425 s. The z-scale is the same for all images. The first and the last frame are measured under ambient conditions and the central frame was obtained during SVA with a 1:4 ratio of toluene and hexane nitrogen vapor. Seven aggregates are marked with white circles. **c** Evolution of $F_{red}$ values with 500 ms time resolution from the seven aggregates (gray curves) and their average (red/orange curve). **d, e** Normalized PL spectra and transient PL decays of a single aggregate in a dry (red) and swollen state (orange). The PL decays are fitted by a single-exponential function with an offset to extract the PL lifetime. **f** Histograms of the full width at half maximum (FWHM) of the PL spectra of 89 swollen PPEB-1 aggregates (orange) and 131 PPEB-1 molecules (blue) (all individual spectra are shown in Supplementary Fig. 1)

excitation energy of 2.82 eV in continuous wave mode. The laser beam is expanded and focused by a lens system onto the back-focal plane of the objective, generating an excitation area of $80 \times 80\,\mu m^2$[37]. The PL is collected by the same objective and split by a dual-view unit into two detection channels—one for PL emission below 2.5 eV (red arrow, Fig. 2a) and the other above 2.5 eV (green arrow, Fig. 2a)—by a dichroic beam splitter as shown in Fig. 2a, and imaged onto different regions of the EMCCD-camera. The two detection channels are shown as green and red in Fig. 2a and are superimposed on top of each other after software corrections for image distortions such as chromatic aberrations.

Such a superimposed image of $40 \times 40\,\mu m^2$ after SVA induced aggregation is shown in the left panel of Fig. 2b, which is an accumulated image of the first 50 s of a 425 s long measurement. During the first 50 s the sample is measured under a nitrogen atmosphere and after 50 s SVA is applied with a toluene/hexane vapor mixture to partially swell the aggregates. The toluene/hexane vapor mixture is chosen because on the one hand toluene is a good solvent for PMMA as well as for PPEB-1, so that the aggregates dissolve during SVA; and on the other hand, hexane is a bad solvent for both compounds and has no significant impact during SVA[36]. The degree of aggregate swelling can therefore be fine-tuned by simply mixing both solvents, and thus at a vapor ratio of 1:4 for toluene and hexane the aggregates do not dissolve. No diffusion or disintegration of the aggregate is discernible on the minute time scale, as is demonstrated by the microscope images in Fig. 2b. The central panel in Fig. 2b shows an accumulated image of the same sample area from time 100 to 250 s, during which the sample is purged with the 1:4 toluene/hexane vapor mixture. After 250 s the SVA is stopped and the sample is subsequently dried under a nitrogen atmosphere. The right panel shows an accumulated image of the same area from time 350 s to 425 s.

The mainly red diffraction-limited spots observed in the left panel of Fig. 2b clearly confirm that most of the aggregates exhibit PL emission below 2.5 eV in the dried state[35]. The central and right panels show that the aggregates do not diffuse during the SVA process with 1:4 toluene/hexane vapor. To confirm this, seven aggregates are marked with white circles in the panels: it is clear that the relative positions between them do not change throughout. However, a distinct change of PL emission is seen for most aggregates during SVA. The marked spots turn green, which implies that the PL emission energy rises significantly above 2.5 eV. This change in PL emission is reversible as can be seen in the right panel, with all marked spots turning red again upon subsequent drying. The fraction of red PL, $F_{red}$, is defined as $I_{red}/(I_{red} + I_{green})$ as shown in Fig. 2a and can be calculated for each aggregate in each frame of the acquisition. Fig. 2c shows $F_{red}$ transients of the seven aggregates for the complete experiment time of 425 s and with time bins of 500 ms. The orange/red curve gives the average of all seven $F_{red}$ transients. Here, the change in PL emission and its reversibility becomes more distinct, with the average of the $F_{red}$ value dropping from 0.85 under ambient conditions to 0.5 during SVA. Thus, we have established a reversible change in PL emission of PPEB-1 aggregates during SVA, which now allows us to compare both spectroscopic states of the aggregates in more detail by using confocal microscopy.

We employ confocal microscopy to obtain PL spectra and transient PL lifetime decays from aggregates before and during SVA[27,37]. The films are scanned by the confocal microscope to identify the positions of the aggregates, which are subsequently placed inside the diffraction-limited excitation spot, where they are excited by a fiber-coupled diode laser (2.82 eV excitation energy, with 20 MHz repetition rate in pulsed mode). The PL is split by a 70/30 beam splitter to simultaneously yield spectra and

transient PL decays by employing time-correlated single-photon counting (TCSPC). A typical spectrum and transient PL decay of a dry aggregate before SVA is shown in Fig. 2d, e as red curves and is compared directly with a swollen aggregate during SVA, shown as orange curves. Both PL decays follow a single-exponential decay with lifetimes $\tau_{PL}$ of 3.51 and 0.47 ns for the dry and swollen aggregate, respectively—a difference of almost an order of magnitude. The dry aggregate exhibits all the characteristics of an aggregate with H-type coupled chromophores[14]: red-shifted PL with a strong decrease of the 0–0 to 0–1 vibronic peak ratio and a long $\tau_{PL}$. In this case, radiation stems from an excimer-like transition, but since the aggregate also exists in the ground state, some residual vibronic structure in the excimer emission is observed[27]. This type of spectrum is strongly broadened and shifted to the red with regards to that of the isolated chains. Moreover, as demonstrated by Walter et al., C=C and C≡C modes in the vibronic progression of the PL can mix with each other, showing up as an additional band in the PL[38]. For this reason, it is not meaningful to extract further information by analyzing vibronic peak ratios at room temperature. The swollen aggregate, on the other hand, exhibits unambiguous characteristics of J-type coupling in the PL[14,20,21]: a narrow 0–0 peak with an increased 0–0 to 0–1 peak ratio and a short $\tau_{PL}$. The swollen aggregate chains show even stronger J-type character compared to single chains as is demonstrated by the full width at half maximum (FWHM) values of the PL spectra of single swollen PPEB-1 aggregates compared to single isolated PPEB-1 molecules, as shown in the histogram in Fig. 2f (all raw data are shown in the Supplementary Fig. 1). The values for single PPEB-1 chains are broadly distributed around a mean of $100 \pm 1$ meV (Fig. 2f, blue bars), whereas for the swollen PPEB-1 aggregates a clear decrease to a mean value of $80 \pm 1$ meV is seen (Fig. 2f, orange bars). We note that the errors regarding the mean FWHM value are the standard error of the mean and not the standard deviation of the distribution. This spectral narrowing can be attributed to improved J-type coupling of the polymer repeat units in the swollen aggregates compared to single chains[39]. According to the model by Knapp, this decrease in linewidth corresponds to an increase in the number of repeat units coupled in the chains in the swollen aggregates compared to isolated chains[40]. Previous work on P3HT nanofibers demonstrated that interchain coupling, i.e., H-type coupling, can also be altered by applying pressure or decreasing the temperature[41,42]. However, we conclude here that the predominant coupling type in mesoscopic aggregates can be discretely switched between H-type and J-type. This result further implies that both coupling mechanisms are present in the dry aggregates. The interchain H-type coupling is completely switched off by partially swelling the aggregates due to the increased distance between neighboring CP chains, leaving behind the intrachain J-type coupling. Surprisingly, we observe that the remaining PL of the aggregated chains shows stronger J-type character compared to single chains. Therefore, small changes in chain morphology can be responsible for large spectroscopic differences: controlling the morphology in the dry state becomes a crucial material parameter.

**Tuning between H-type and J-type coupling by side chain engineering.** We tackle this challenge by modifying the side chains of our model system, introducing the CPs PPEB-2 ($M_n = 66$ kDa and PDI = 1.05) bearing 2-ethylhexyloxy side chains, and PPEB-3 ($M_n = 78$ kDa and PDI = 1.07) with cholestenol-substituted benzyloxy side chains, both shown in Fig. 1c. Single chains and isolated aggregates thereof prepared by SVA are measured using the confocal microscope setup. A first impression is acquired by measuring several hundred single particle spectra

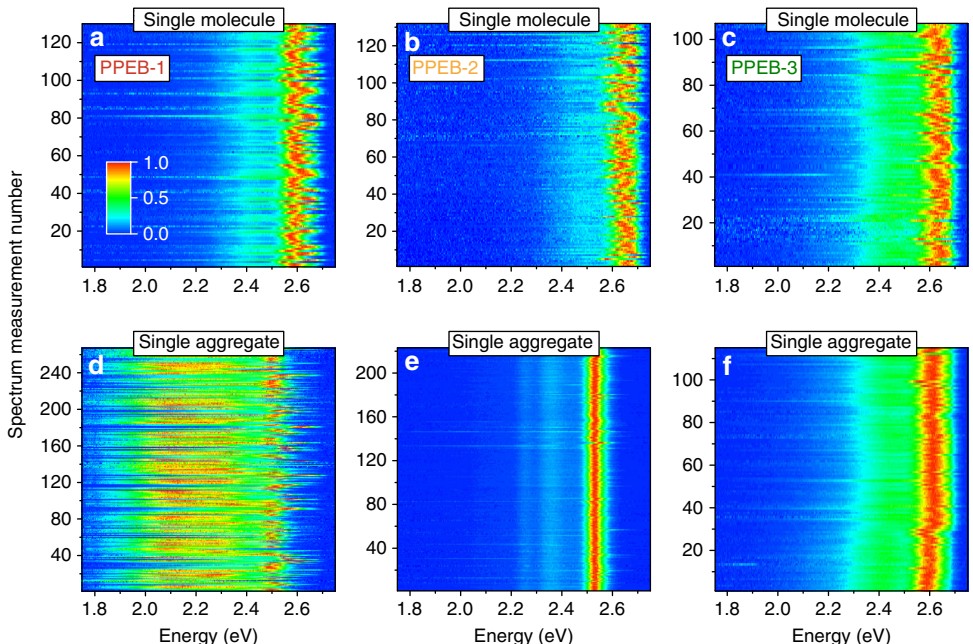

**Fig. 3** Comparison of single-molecule and single-aggregate spectra. **a–c** Normalized single-molecule spectra plotted on a false-color intensity scale of PPEB-1 **a**, PPEB-2 **b**, and PPEB-3 **c**. **d–f** Single-aggregate spectra of PPEB-1 **d**, PPEB-2 **e**, and PPEB-3 **f**. Aggregates were grown by in situ SVA, subsequently dried under nitrogen and measured in air

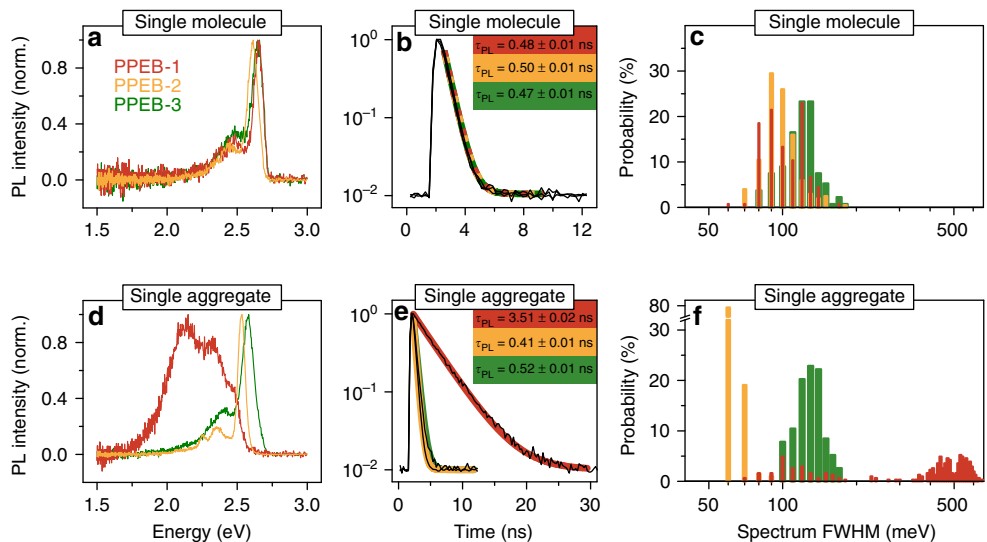

**Fig. 4** PL spectra and PL decays of single chains and single aggregates. Normalized PL spectra, transient PL decays, and spectral FWHM values extracted from the spectra in Fig. 3 and plotted in a histogram with 10 meV bin size for single molecules **a–c** and single aggregates **d–f** of PPEB-1 (red), PPEB-2 (orange), and PPEB-3 (green) in PMMA measured in air. Transient PL decays were fitted with an exponential plus an offset to account for the background, yielding the PL lifetimes stated in **b** and **e**

of PPEB-1, PPEB-2, and PPEB-3 single chains as well as aggregates thereof. We plot these normalized spectra in Fig. 3 on a false-color scale, with the *y*-axis representing the arbitrary measurement index number.

Irrespective of the material, all single-molecule spectra exhibit a pronounced 0–0 transition at ~2.6 eV, which scatters from molecule to molecule from 2.55 eV up to 2.65 eV, due to slight changes in the dielectric environment[43]. A difference is seen in the case of single-molecule PPEB-3 spectra in panel c, which exhibit a somewhat stronger PL intensity in the vibronic progression at ~2.4 eV compared to PPEB-1 and PPEB-2. This effect can be attributed to a slightly higher degree of disorder in

single PPEB-3 chains owing to the large bulky side chains in this material. The three polymers are aggregated by the same SVA procedure, as described in the methods section, yielding similar dry aggregates with respect to their average size. In situ SVA enables us to count the average number of single chains and aggregates per fluorescence microscope image before and after SVA, providing the average number of chains within an aggregate[33,35]: here, on average ~15 chains make up an aggregate for all samples shown. Panels d–f demonstrate the dramatic impact of the different side chains on the single aggregate spectra: PPEB-1 aggregates exhibit a slight red-shift of the narrow 0–0 transition to 2.5 eV and most of the spectra display a strong

increase of the vibronic progression between 2.4 and 1.9 eV. PPEB-2 aggregates also show a red-shift of the 0–0 transition down to ~2.52 eV, but a much less broad vibronic emission. The vibronic progression of the C=C and C≡C bands at ~2.32 and 2.23 eV can be resolved despite the low intensity, thanks to the narrow transition line widths. PPEB-3 aggregate spectra are not discernible from their single-molecule counterparts. Here, it is important to note, that even the 0–0 transition does not change when going from single chains to aggregates, indicating that solid-state solvato-chromism—spectral red-shifting upon aggregation due to the different dielectric environment experienced by single chains in PMMA and in PPEB-3 aggregates—only has a minor impact.

For further comparison, we show typical single-chain and aggregate spectra with corresponding transient PL decays and extract FWHM values of all spectra to plot them in a histogram in Fig. 4 for the three compounds. The single-molecule spectra and $\tau_{PL}$ are virtually indistinguishable from each other for the three materials (panel a) with $\tau_{PL}$ values between 0.47 and 0.5 ns (panel b). In addition, the spectral widths for all three compounds at the single-molecule level are very similar and range between 80 and 150 meV (panel c). The different side chains therefore do not affect the photophysics of this material at the single-molecule level.

However, the different side chains do play a major role regarding the photophysics of isolated aggregates, as shown in Figs 3d–f and 4d–f. By changing the side chains from hexyloxy to 2-ethylhexyloxy, the dominant coupling switches from H-type to J-type. The single-aggregate spectrum and transient PL decay of PPEB-2 shows all expected characteristics for such J-type coupling: a narrow, red-shifted 0–0 peak, increased 0–0 to 0–1 peak ratio, and a shortened $\tau_{PL}$ of 0.41 ns. The spectral widths of PPEB-2 aggregates are on average only $70 \pm 1$ meV (Fig. 4f, orange bars) as compared to $100 \pm 1$ meV for PPEB-2 single chains. By invoking the model of Knapp again, this difference in linewidth corresponds to an increase in the number of repeat units coupled along a single chain in the aggregate compared to the isolated single chain[40]. This effect is also referred to as aggregation-induced planarization, and has been demonstrated for different conjugated oligomers and polymers[44,45], most prominently polyfluorene[46]. The torsional angles between the monomers may change upon SVA and aggregation. A well-known example of this effect is found in polyfluorene, which can transition from the twisted glassy phase to the planarized beta-phase under SVA and gives rise to a dramatic change in vibrational modes in cryogenic single-chain PL spectra[47]. We expect that a similar effect will arise in PPEB aggregates, but testing this will require combining the SVA technique with cryogenic SMS. The distribution of spectral widths for the PPEB-1 aggregates (Fig. 4f, red bars) exhibits three distinct populations, due to different degrees of coupling: first, widths of 80–150 meV are found from spectra with dominant 0–0 transitions, a spectral signature, which corresponds to the case of weak H-type coupling; second, 400–500 meV spectral widths are extracted from spectra with a dominant vibronic progression, which corresponds to strong H-type coupling; and, third, values of 500–600 meV are determined from spectra with equal contributions of the 0–0 transition and the vibronic progression, a situation which can be referred to as intermediate H-type coupling strength. In contrast, comparison of the PPEB-3 single-chain (Fig. 4a–c, green data) and aggregate (Fig. 4d–f, green data) results demonstrates that the bulky side chains of PPEB-3 prevent any kind of electronic coupling as is evidenced by the absence of any differences between these data sets.

To obtain an overview regarding the heterogeneity of the samples, we next measure $\tau_{PL}$, $F_{red}$ values, and the PL intensities

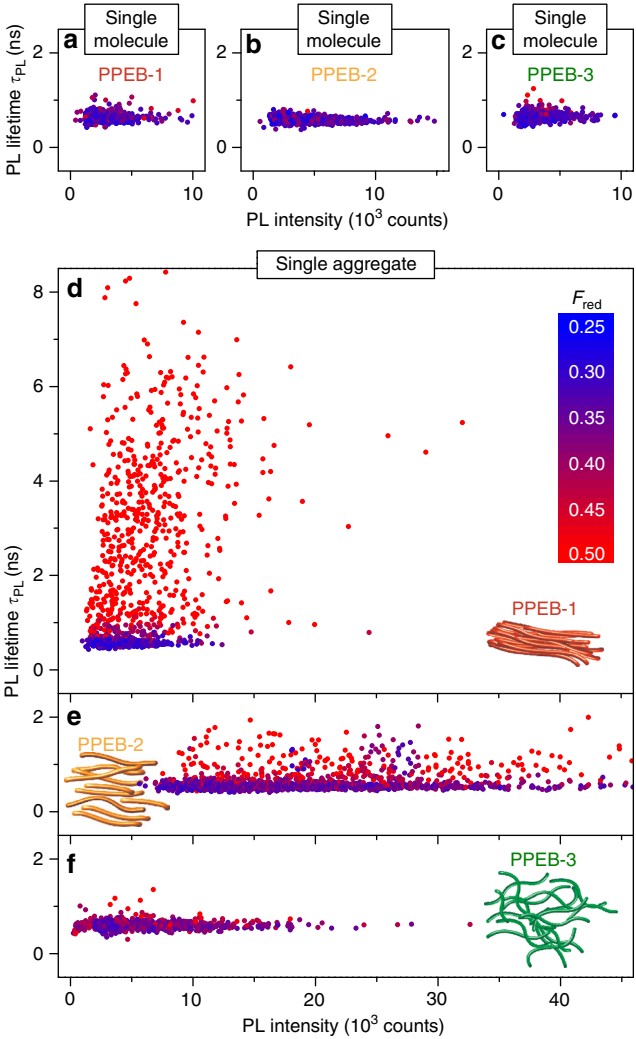

**Fig. 5** Comparison of the statistics of single chains and aggregate samples. **a–c** Scatter plots of PL lifetime, $\tau_{PL}$, and PL intensity for single chains and isolated aggregates **d–f** formed by SVA of the three materials measured in air. The color encodes the fraction of red emission, $F_{red}$, for each measured single particle, which ranges between 0.25 (blue) and 0.5 (red). Data from 391 **a**, 694 **b**, and 469 **c** single chains and 1196 **d**, 1881 **e**, and 889 **f** isolated aggregates are shown

of a large number of particles with confocal scanning microscopy and plot the scatter of these in Fig. 5. Six different samples are compared: three single-molecule samples and three aggregate samples of PPEB-1, PPEB-2, and PPEB-3, respectively. For all samples $\tau_{PL}$ is plotted against average PL intensity. The color of the data points encodes the $F_{red}$ value, ranging from 0.25 to 0.5. The scaling is equal for all panels to allow direct comparison of the distributions. We note that the size and therefore the absorption cross-section distribution of the PPEB-1 and PPEB-2 aggregate samples are comparable, so that PL intensity variations can be linked directly to the differences in PL quantum yields.

The distribution of all values is very narrow and almost the same for the three single-molecule samples, as seen in Fig. 5a–c. The PL intensity ranges between almost 0 and 10,000 counts, $\tau_{PL}$ is between 0.4 and 1 ns and $F_{red}$ between 0.25 and 0.35 for all single-molecule samples. These observations confirm that the side chains do not influence the photophysics of the PPEB backbone at the single-molecule level. This conclusion is in contrast to the single aggregates shown in Fig. 5d–f. A large heterogeneity of $\tau_{PL}$ is present ranging from 0.5 to 8 ns, which coincides with a

spectral red-shift demonstrated by an increase in $F_{red}$ to > 0.5 for aggregates consisting of PPEB-1. At the same time, the average PL intensity rises only slightly compared to the single-molecule sample, even though the fluorescent spots measured in the microscope now comprise, on average, ~15 single chains. These observations can again be rationalized by H-type coupling occurring in PPEB-1 aggregates. The transition from the lowest excited state to the ground state becomes dipole forbidden, hence the increase of $\tau_{PL}$ and the overall integrated red-shifted PL, which is excimeric in nature as previously discussed for model systems of this material[27]. The large scatter of $\tau_{PL}$ can be attributed to small variations in interchain separation in the aggregates, which leads to different interchromophoric coupling strengths[21,27]. The stronger the coupling, the longer the lifetime, the more red-shifted the emission—hence the observed correlation. The average PL intensity does not increase despite the fact that the aggregates consist of multiple single chains because the radiative rate is reduced by H-type coupling. With an intrinsic non-radiative decay being present in this material, a reduction of PL quantum yield will therefore arise for H-type coupled aggregates[27]. PPEB-2 aggregates exhibit a much narrower distribution of $\tau_{PL}$, with no values measured above 2 ns. However, a similar correlation between $F_{red}$ and $\tau_{PL}$ to that found in PPEB-1 aggregates is seen in Fig. 5e, indicating that H-type coupling still occurs but is less dominant. However, H-type coupling does not swamp the intrachain J-type coupling in this case, as is evidenced by the increased average PL intensity with respect to PPEB-1 aggregates and the fact that the 0–0 transitions remain rather narrow (see Fig. 3e). The PL quantum yield of PPEB-2 aggregates is therefore higher compared to that of PPEB-1 aggregates. It is important to note that the acquisition method for measuring $F_{red}$, $\tau_{PL}$, and PL intensity of a diffraction-limited spot is different compared to measuring spectra of single particle. For the first method, only confocal scan images are obtained in which the particles are illuminated for ~100 ms, whereas for the second an integration time of ~4 s per spectrum is needed. This increase in illumination time of the emitters can lead to subtle differences due to photo selection and photo degradation of the emitter. H-type emitters within the PPEB-2 aggregates may bleach with time, leaving behind J-type emitters, which in turn may exhibit a higher degree of photo-stability compared to the H-type emitters. The longer excited-state lifetime of H-type emitters likely provides a higher probability to undergo irreversible photochemical reactions. Finally, in Fig. 5f we contrast these results with aggregates grown of PPEB-3, which demonstrates very similar distributions of spectroscopic observables to that of the single-molecule samples. Only slightly higher PL intensity values are measured for PPEB-3 aggregates suggesting a larger absorption cross-section of the aggregates compared to the single chains. Due to the bulky side chains, the solubility of PPEB-3 is higher compared to that of PPEB-1 and PPEB-2, which will lead to a different size distribution of the aggregates and a higher number of "left-over" single chains[33], which do not aggregate. These single PPEB-3 chains cannot be distinguished spectrally from aggregates and appear in the scatter plot at lower PL intensity values. However, no correlation between $\tau_{PL}$ and $F_{red}$ is apparent for these aggregates, suggesting negligible interchromophoric coupling. We conclude that the large cholestenol benzyloxy side chains prevent interchain H-type coupling as well as intrachain J-type coupling.

**Morphology of J-type and H-type coupled CP aggregates**. A connection between the electronic coupling and the nanoscale morphology is anticipated as formulated in the introductory section. To assess the morphology, we measure the internal degree of order in single-chain and single-aggregate samples by employing excitation polarization fluorescence spectroscopy to establish a connection with the dominant form of electronic coupling[48–50].

The samples are investigated by widefield rather than confocal fluorescence microscopy to obtain the PL intensity dependence with respect to the polarization of the excitation beam as sketched in Fig. 6a. In this way, many single objects can be studied at once. The modulation of PL intensity reports on the overall anisotropy of the molecular object in absorption. The polarization of the excitation beam is rotated in the xy-plane of the sample and the PL intensity, I, is recorded simultaneously. This quantity can be described by Malus' law,

$$I(\theta) \propto 1 + M cos2(\theta - \Phi), \qquad (1)$$

where $M$ is the excitation polarization-modulation depth, $\theta$ is the angle of polarization of the incident light, and $\Phi$ corresponds to the orientation angle of the overall molecular TDM for maximal PL. $M$ is measured for each single diffraction-limited spot and is summarized in Fig. 6b, c in histograms with bins of 0.1 for single chains and aggregates of the three samples. For direct comparison, the histograms are normalized to the overall number of measured spots for each sample. The probability of measuring a certain $M$ value is given on the y-axis. The single-molecule samples exhibit similar broad distributions of $M$ peaking at 0.4–0.5, shown in Fig. 6b, suggesting that the different side chains have no impact on the morphology at the single-molecule level. However, as for the spectroscopic properties discussed above, the chain morphology is very different for the aggregated samples: the resulting $M$ histogram of PPEB-1 aggregates has a maximum at $M \sim 0.7$ (Fig. 6c, red), which suggests well-ordered aggregates, whereas PPEB-2 aggregates have the highest probability of measuring $M$ values around 0.2–0.3 (Fig. 6c, orange), implying a higher degree of disorder. PPEB-3 aggregates are effectively completely disordered, with $M$ peaking at 0.1 with a narrow distribution (Fig. 6c, green). Thus, depending on the side chains, the degree of aggregate order differs significantly. We conclude that it is the chain morphology in the aggregate which is primarily responsible for the different types of electronic coupling found in the aggregates. For H-type coupling, intrachain and interchain ordering is necessary, which in turn leads to a high degree of overall TDM alignment (red bars). In contrast, for J-type coupling, intrachain ordering is sufficient, whereas H-type coupling is removed entirely if the aggregate becomes disordered (yellow bars). In the disordered PPEB-3 aggregates (green bars) H-type coupling is suppressed as is any additional J-type coupling. We note that it is well-known that intrachain conformational disorder can substantially reduce the efficiency of interchain energy transfer in bulk films[51]. As discussed below, this reduction in interchromophoric coupling by Förster-type incoherent dipole-dipole coupling is clearly manifested in the photon statistics of the single-aggregate fluorescence.

Consequently, the ordering and the resulting type of interchromophoric coupling will also impact the overall amount of light-absorbing and emitting material which can couple together in the mesoscopic aggregate to form a single quantum emitter. It is expected that the high degree of order in H-type coupled aggregates, i.e., the coexistence of both interchain and intrachain order, together with the increased excited state lifetimes, will promote interchain energy transfer in CPs[52]. In contrast, in J-type aggregates, it will be predominantly the material of one and the same CP chain where interactions and energy transfer occur. A signature of electronic interactions within the material is the degree of single-photon emission stemming from such aggregates as a consequence of singlet-singlet annihilation (SSA)[53,54]. SSA is a result of

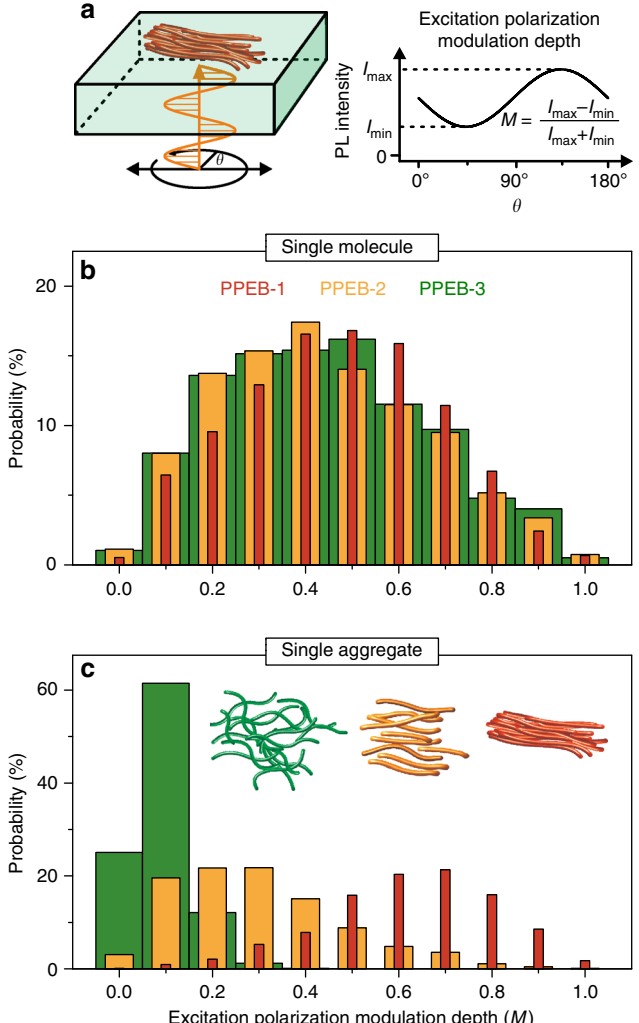

**Fig. 6** Morphology of single chains and isolated aggregates. **a** Sketch of the excitation polarization measurement technique and definition of the modulation depth, $M$. **b** $M$ values for single chains and isolated aggregates **c** of the three samples measured in air. The spacing between the bars is varied for clarity and the histograms are normalized to the overall number of measurements (single chains: 1097, 1766, 977; aggregates: 743, 1062, 772) of spots yielding the probability of measuring a particular $M$ value

interchromophoric interactions by dipole-dipole coupling, either coherent or incoherent transfer of excitation energy, within the multichromophoric system[55]. A common mechanism of incoherent coupling is fluorescence resonance energy transfer.

**Assessing interchromophoric interactions from fluorescence-photon statistics.** We measure the second-order cross-correlation function $g^{(2)}(\Delta\tau) = \langle I(t) \cdot I(t + \Delta\tau) \rangle / \langle I(t) \rangle^2$ by splitting the detection path of the confocal microscope equally onto two detectors as shown in the inset of Fig. 7a. The photon statistics are obtained under ambient conditions and with pulsed excitation at a repetition rate of 10 MHz. Only the first second of the acquisition period is taken into account to minimize photo bleaching and blinking effects, which can reduce the quality of photon antibunching[56,57]. We measure multiple single aggregates of each material to obtain sufficient photon statistics for the correlation plot. The correlation value at zero time delay between the detectors, $g^{(2)}(0)$, can be translated into the average number, $N$, of independently emitting single-photon sources by $N = 1/(1 - g^{(2)}(0))$[58,59]. Such a single emissive unit can only

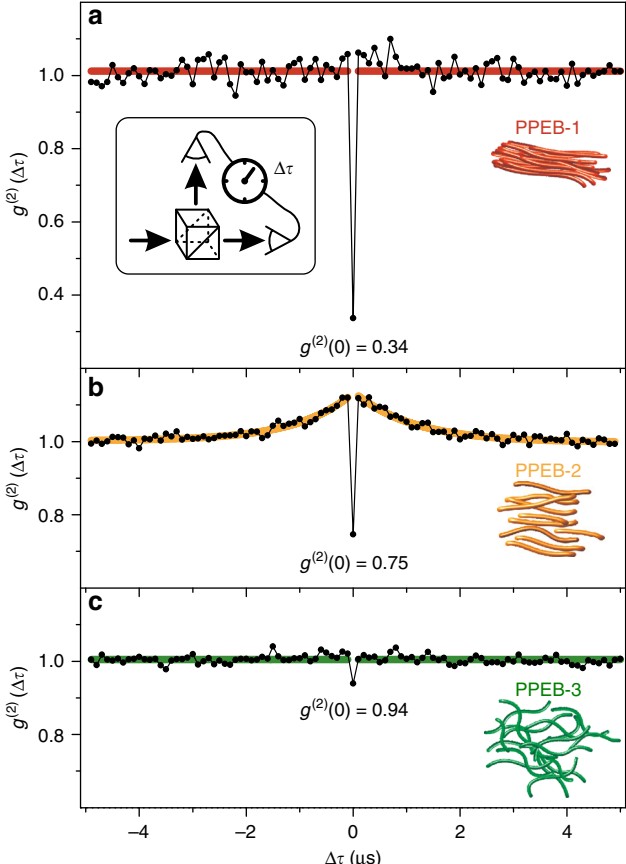

**Fig. 7** Photon statistics of single multichain aggregates measured in air. Accumulated second-order cross-correlation, $g^{(2)}(\Delta\tau)$, curves for 57 PPEB-1 **a**, 47 PPEB-2 **b**, and 44 PPEB-3 **c** aggregates. The aggregates were excited by laser pulses (10 MHz repetition rate) to control the difference $\Delta\tau$ in photon arrival time between the two detectors. The antibunching dip at $\Delta\tau = 0$ is apparent. In **b**, photon bunching occurs for $|\Delta\tau| > 0$, which is described by a single-exponential fit (orange)

emit one single photon during its excited state lifetime and may consist of either a single chromophore or multiple chromophores coupled together by coherent or incoherent energy transfer. The accumulated $g^{(2)}(\Delta\tau)$ is plotted for 57 PPEB-1 aggregates in Fig. 7a. The red line is a guide to the eye and demonstrates that no photon bunching occurs on the microsecond time scale (i.e., $g^{(2)}(\Delta\tau \neq 0) \approx 1$), implying the absence of long-lived dark states such as a triplet or a charge transfer (CT) state[60]. However, strong photon antibunching arises in the PPEB-1 aggregates, with $g^{(2)}(0) = 0.34$, which renders $N$ between one and two on average for all measured PPEB-1 aggregates. The 47 measured J-type PPEB-2 aggregates display a strikingly different behavior regarding the photon statistics, plotted in Fig. 7b. First, photon bunching is clearly present on the microsecond time scale. This bunching is described by a single-exponential fit (orange curve) to the correlation function, and relates to the formation of a meta-stable dark state, such as a CT state or a molecular triplet. These states interact with the fluorescing singlets through singlet-polaron or singlet-triplet quenching[61]. Recent work on poly(3-hexylthiophene) (P3HT) nanofibers, single chains, and isolated mesoscopic aggregates also suggests that triplet states are more easily formed in J-type than in H-type aggregates, but the mechanism behind this phenomenon is still an open question[62,63]. We note that the oxygen in the ambient air should quench triplet dark states, but this effect may be impeded by deeply embedded aggregates in PMMA[64]. Second, photon

antibunching clearly still arises in PPEB-2 aggregates, but its quality is substantially reduced compared to the H-type aggregates, with $g^{(2)}(0) = 0.75$, leading to $N \approx 4$. Finally, the 44 measured uncoupled PPEB-3 aggregates exhibit no photon bunching at all (Fig. 7c, green line) and virtually no photon antibunching, with $g^{(2)}(0) = 0.94$, implying $N \geq 17$ independently emitting units. Together with the information that the PPEB-1 and PPEB-2 aggregates likely contain more chains than the PPEB-3 aggregates, because of the greater solubility of PPEB-3, we draw the conclusion that *at least* 17 distinct emitting units within the aggregate couple together to behave as, on average, 1–2 units in the H-type aggregates and as ~4 units in the J-type aggregates of the polymer. This conclusion confirms the hypothesis that H-type coupling leads to an overall collection of more interacting material as compared to J-type coupling, an effect which is a simple consequence of the closer proximity of the π-conjugated segments in PPEB-1. We note that our study probes the semantic limits of CP photophysics. While the concept of a chromophore on a CP chain is, in principle, well defined as a finite number of repeat units[17], which give rise to universal spectral features at cryogenic temperatures[31,65], the term "chromophore" really implies the region of the material, which lends it a particular color. As such, the excimer-like emission of the H-aggregate can also be attributed to a "chromophore", showing photon antibunching. In this case, however, the emitting region actually consists of multiple CP chromophores, which couple together coherently to form one emitting center.

## Discussion

This work exemplifies the strength of combining single-molecule and single-aggregate spectroscopy in materials science to resolve the differences between single CP chains and aggregates thereof. No differences are detectable at the single-molecule level between the three PPEB samples. Only single-aggregate spectroscopy reveals the impact of the different side chains on coherent coupling and the relationship with chain morphology, interchromophoric energy-transfer properties and subsequent CT state formation. The techniques developed for SMS are directly applicable to aggregates. The fine control of SVA is an important method for studying isolated aggregates. First, we establish that careful swelling by SVA of PPEB-1 aggregates enables the reversible switching between H-type and J-type coupling by selectively deactivating H-type coupling. This deactivation is achieved by prying the chains apart slightly by incorporating solvent molecules through SVA. We conclude that intrachain J-type coupling is always present in such aggregates, but is completely masked in terms of its spectral signatures by interchain H-type coupling. Second, by fine tuning the SVA aggregation process, isolated aggregates of different samples but of comparable sizes can be grown. This tool offers the possibility to compare spectroscopic observables and draw conclusions regarding the impact of different electronic coupling types as well as their morphological prerequisites. The side chains differ only marginally between PPEB-1 and PPEB-2 (hexyloxy and 2-ethylhexyloxy, respectively), but the impact on coupling in aggregates is dramatic. Electronic coupling can only be switched off by bulky side chains in PPEB-3. The degree of morphological order is highest in H-type aggregates as is the funneling of excitation energy to a single emissive unit. This effect is the cause of the strong photon antibunching in the multichain aggregate. This funneling decreases in less-ordered J-type aggregates. However, a dark state evolves in J-type aggregates, manifested in photon bunching, which is an indication of an increased yield of triplet or CT state formation. Presumably, these dark states are quenched by the interaction between chromophores in multichain aggregates, in analogy to the strong quenching of triplets seen in P3HT aggregates and films[63,66]. Our findings provide new insights and illustrate sensitive measurement techniques for the development of the design principles for organic electronics, in particular with regards to controlling the form and degree of electronic coupling between excited states, with interesting consequences regarding excitation energy diffusion, dark-state generation, and charge separation.

## Methods

**Sample fabrication.** Poly(*para*-phenylene-ethynylene-butadiynylene) with hexyloxy side chains (PPEB-1), 2-ethylhexyloxy (PPEB-2), and cholestenol-substituted benzyloxy side chains (PPEB-3) were synthesized as described elsewhere[67] and in detail in the Supplementary Note 1. The samples were purified using a gel-permeation chromatograph (GPC) to obtain three samples with a number-average molecular weight $M_n = 40$ kDa with a PDI of 1.46 of PPEB-1, $M_n = 66$ kDa with a PDI of 1.05 of PPEB-2 and $M_n = 78$ kDa with a PDI of 1.07 of PPEB-3. Poly (methyl-methacrylate) (PMMA, $M_n = 46$ kDa, PDI = 2.2) was purchased from Sigma-Aldrich. Isolated chains of PPEB molecules were embedded in a PMMA host matrix by dynamically spin-coating from toluene on glass cover slips, which were cleaned according to a published procedure[37]. The PMMA film thickness was 200–250 nm, and the concentration of PPEB in solution before spin-coating was ~$10^{-10}$ mol•l$^{-1}$ and ~$10^{-9}$ mol•l$^{-1}$ for the single-molecule and aggregate samples, respectively. The samples were incorporated into a gas flow cell and annealed under solvent vapor for 20 min with a fixed acetone/chloroform ratio of 1:4 to prepare similarly sized aggregates. The acetone/chloroform vapor was prepared by purging nitrogen through two solvent reservoirs containing 100 ml of dry acetone and chloroform, respectively. The gas flow was controlled by mass flow controllers (MKS Instruments) and adjusted to be 8 sccm (standard cubic centimetres per minute) and 2 sccm for acetone and chloroform, respectively. The acetone and chloroform saturated nitrogen vapor streams were combined and used to swell the sample. After 20 min of SVA the samples were dried under pure nitrogen for 5 min. For partly swelling the PPEB-1 aggregates toluene and hexane was used with the same procedure and a ratio of 1:4. All samples were measured in air to quench meta-stable dark states, such as triplet states, except for measurements on swollen aggregates, which were conducted under a constant nitrogen flow for solvent vapor annealing.

**Dual-view wide-field fluorescence microscopy.** An inverted microscope (Olympus IX71) was used for wide-field as well as confocal excitation and detection. The excitation source was provided by a fiber-coupled diode laser (PicoQuant, LDH-C-440) with a wavelength of 440 nm in continuous wave mode and the excitation light was passed through a clean-up filter (AHF analysis technology, HC Laser Clean-up MaxDiode 445/10). The laser beam was expanded and focused via a lens system onto the back-focal plane of the 1.35 NA oil-immersion objective (Olympus, UPLSAPO 60XO) through the back port and a dichroic mirror (AHF analysis technique RDC 442 nt) in the microscope. An excitation area of ~$80 \times 80$ µm² was generated in the focal plane and the fluorescence of the sample is collected by the same objective and passes through the dichroic mirror. The fluorescence signal was imaged on an EMCCD camera (Andor, iXon3 897) after an additional magnification of 1.6 times and after passing a dual-view unit (CAIRNS OptoSplit 2) to generate two equal detection areas, which are sensitive to different emission wavelengths. The dual-view unit was equipped with a dichroic beam splitter (AHF analysis technology, HC BS 495) to differentiate between PL below and above 2.5 eV. Images were analyzed with a home-written software using LabView.

**Excitation polarization spectroscopy.** The same microscope was used in wide-field excitation mode for the excitation polarization measurements, but without the dual-view unit in front of the camera. Details can be found in ref. [37]. In brief, a Glan-Thompson polarizer was inserted in the excitation path to provide linearly polarized excitation light. The polarization of the excitation light was rotated by an electro-optical modulator (FastPulse Technology Inc., 3079-4PW) and an additional λ/4 waveplate, as described elsewhere[68]. The excitation intensity was set to 100 mW/cm² and the overall magnification resulted in a resolution of 160 nm² per pixel leading to diffraction-limited spots of ~$2 \times 2$ pixels for a single molecule or aggregate. The polarization of the excitation light was rotated in the $x$–$y$-plane by 180° at periods of 20 s and the fluorescence intensity of each individual fluorescent spot was recorded as a function of the polarization angle. A point size of $5 \times 5$ pixels was assumed for the calculation of the overall intensity and the local background of the surrounding area ($13 \times 13$ pixels) was subtracted for each molecule. The data analysis was conducted with a home-written software provided by the lab of the late Paul Barbara using MATLAB[69].

**Scanning confocal microscopy.** The samples were investigated with a scanning confocal microscope based on an Olympus IX71[37]. Excitation was carried out by a fiber-coupled diode laser (PicoQuant, LDH-D-C-440) at 440 nm under pulsed excitation with a repetition rate of 10 MHz for photon statistics measurements or

20 and 80 MHz for PL lifetime measurements. The excitation light was expanded and collimated via a lens system to a beam diameter of ~1 cm and coupled into the oil-immersion objective for confocal excitation with an intensity set to 50 W/cm². Fluorescence images (size of $20 \times 20 \, \mu m^2$, integration time 2 ms/pixel with a resolution of 50 nm/pixel) were recorded by stage scanning (Physik Instrumente, model P-527.3CL). The fluorescence signal passed a 50 μm pinhole and fluorescence filter (AHF Analysentechnik, Edge Basic LP 442 long pass filter) and was split by a dichroic mirror (AHF Analysentechnik, HC BS 495) and detected by two avalanche photodiodes (APDs, PicoQuant, τ-SPAD-20) connected to a time-correlated single-photon counting module (TCSPC, PicoQuant GmbH. HydraHarp 400) for extracting the fraction of red emission, $F_{red}$. The images were evaluated by a home-written LabView software capable of automatically detecting single spots for which $F_{red}$ was calculated, and simultaneously the PL lifetime and PL intensity were extracted. Alternatively, the fluorescence signal was split by a 70/30 beam splitter to simultaneously detect 30% of the PL on an avalanche photodiode (Micro Photon Devices S.r.l., PDM Series) connected to the TCSPC unit, and 70% on a spectrograph (Andor technology plc., SR-303i-B) coupled with a CCD camera (Andor Technology plc., DU401A-BV) to obtain PL decays and spectra from spots which were subsequently placed inside the excitation focus. For photon statistics measurements the fluorescence signal was split by a 50/50 beam splitter and detected by two avalanche photodiodes both connected to the TCSPC unit to record time-tagged photon arrival times, which were further analyzed by a Lab-View program.

**Data availability**. All relevant data are available from the authors.

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

## Acknowledgements

We are indebted to the European Research Council for funding through the Starting Grant MolMesON (305020), the Deutsche Forschungsgemeinschaft for support through the GRK 1570 and to the Volkswagen Foundation for continued support of the collaboration.

## Author contributions

T.E., T.S., M.G. and J.V. conceived, designed and performed experiments, and analyzed the data. K.R., D.L. and S.H. designed and synthesized the compounds. J.V. and J.M.L. wrote the manuscript.

## Additional information

**Competing interests:** The authors declare no competing financial interests.

