## [Peer Review File · Nature Communications]

Reviewers' comments:

Reviewer #2 (Remarks to the Author):

This is a fabulous paper, which demonstrates through carefully designed experiments the ability to tune between J and H-aggregates in conjugated polymers. Such polymers are technologically of great importance in a range of applications from light emitting diodes to solar cells.

The authors use a solvent swelling technique to modulate the packing distance between polymer chains within a given aggregate, effectively turning on and off the interchain coupling. This has a profound impact on the photo physical properties; in the emitting state of H-aggregates the chain dipoles are out of phase leading to a coherent suppression of the radiative decay rate and the 0-0/0-1 PL ratio. By contrast, isolated chains behave as J-aggregates where exciton coherence along the chain leads to enhanced radiative decay rates and 0-0/0-1 ratios. The experiments beautifully bear out these behaviors in quite dramatic fashion; the fluorescence decay rate and PL ratio is enhanced by an order of magnitude in J vs H-aggregates. The authors go on to further demonstrate H/J tuning via side-chain engineering. In addition, they show superior chain alignment within H-aggregates via polarization resolved single-aggregate PL measurements and utilize photon correlation techniques to determine the effective number of emitters. All of these experiments clearly reveal the intricate morphology-dependent photophysics at play in conjugated polymers. This paper will certainly appeal to a broad range of scientists interested in energy transport in organic materials and will no doubt generate considerable excitement especially with regard to the impact of coherence, which is also suspected of enhancing function in biological materials.

There is only one issue that needs to be addressed. On the first page the authors claim that the J-aggregate behavior of isolated chains arises from a head-to-tail arrangement of transition dipole moments a la Kasha theory. In fact, the main reason single polymer chains behave as J-aggregates is due to the strong covalent coupling between repeat units. Conjugated polymers behave as direct band-gap semiconductors: the opposite curvatures of the conduction and valence bands gives rise to an exciton band which has positive curvature at $k=0$, which is the requirement for J-type photophysics. This is described in the paper,

"Strong Photophysical Similarities between Conjugated Polymers and J-aggregates", J. Phys. Chem. Lett. 5, 622 2014

The positive curvature is directly linked to an effective coupling between neighboring repeat units which is negative(see Eq. 13) , as exists in J-aggregates. (There may be also contributions from the interacting transition dipoles which will enforce the J-like behavior of CPs if they are arranged in a head to tail fashion).

Hence, I would recommend that the authors slightly rephrase the introduction to reflect the comments above.

Since exciton coherence is an important theme in this paper, the authors may also want to mention (but this is not required) that for J-aggregates, like single-chain CPs, the PL ratio is a direct measure of the number of coherently coupled repeat units. Hence, the ratio grows with coherence length, as shown in J. Chem. Phys. 135, 054906 (2011) for CPs. The authors may want to reference some of Michel Schott's papers where he is able to grow effectively defect-free straight polydiacetylene chains which have a huge 0-0/0-1 ratio of about 100 (!) consistent with a very long coherence length of at least hundreds of nanometers, see

Dubin, R. Melet, T. Barisien, R. Grousson, L. Legrand, M. Schott, and V. Voliotist, *Nat. Phys.* 2(1), 32 (2006).

Lecuiller, J. Berrehar, J. D. Ganiere, C. Lapersonne-Meyer, P. Lavallard, and M. Schott, *Phys Rev. B* 66(12), 125205 (2002)

Reviewer #3 (Remarks to the Author):

Eder et al. use single-molecule spectroscopy to study the conjugated polymer poly(p-phenylene-ethynylene) with different side groups that affect its aggregation behaviour. Controlled aggregation was achieved by solvent-vapour annealing. The aggregation, or more specifically, the electronic coupling between chains was investigated by PL spectroscopy, lifetime measurements, excitation polarisation spectroscopy, and photon antibunching experiments on isolated aggregates / single chains. Extensive data sets were acquired comprising several hundreds of chains/aggregates.

However, there are several issues, in particular with the interpretation of data, that prevent publication in present form:

1. The expression "coherent coupling mode" is very unusual (I never came across that so far). Either H-type or J-type coupling is meant. In both cases the interaction is the same (Coulomb-coupling), there is no fundamentally different physics involved. Please stick to the common notation in the field.

2. Related to point 1 above: The authors refer to intra-chain (J-type) and inter-chain (H-type) coupling between "neighbouring chromophoric units" (l. 39); and "J-type coupling occurs predominantly between chromophoric units on the same CP chain ..." (l. 42/43). Here I miss clear definitions of "chromophoric units". In this manuscript, intra-chain J-type coupling appears to me to be the interaction between chemical repeating units of a chain, and NOT interactions between chromophores of a chain (if I do understand the sketches in Fig. 1a correctly). It is worth reading Spano's paper on HJ-aggregates carefully (*J. Chem. Phys.* 2013). He defines precisely what is interacting with what on a chain and between different chains. The introduction and several parts of the manuscript have to be clarified / rewritten in this respect.

3. In lines 92-94 the authors claim that switching between H- and J-type coupling was only predicted so far (which again requires a clear definition of those interactions). However, such behaviour was already observed for P3HT-Nanofibres by Grey et al. (*J. Phys. Chem. B* 2012, *J. Phys. Chem. Lett.* 2012), and to some extent for for MEH-PPV by A. Köhler et al. (*J. Chem. Phys.* 2013).

4. The "dry" PL spectra of PPEB-1 in Fig. 2d possess a very strange shape, which is not consistent with the emission of a single H-type aggregate although the highest energy peak is indeed suppressed; yet, the 0-1/0-2 ratio does not fit. This spectrum rather appears to be a superposition of several H-aggregates emitting at different spectral positions, consistent with on average 1-2 emitters shown by antibunching experiments.

5. Line 194: "Narrow 0-0 peaks" from "J-type coupled chromophores" are mentioned. What is narrow? Narrow compared to what? Please quantify. Moreover, the "swollen" PPEB-1 aggregate spectrum looks very similar to that of a single PPEB-1 chain, the lifetimes are essentially identical (cf. Fig. 2d,e and 3a,b). On which basis is the "swollen" spectrum then assigned to that of a J-aggregate? The "swollen" chains might be more planar than a single isolated chain (e.g. P3HT planarises before it forms H-aggregates). But why is that potentially more planar chain then a J-aggregate (Fig. 2d), for which the authors claim that several "chromophoric units" along the same chain interact? Why is the single chain

then not a J-aggregate (Fig. 3a)? This discrimination makes no sense based on such very minor spectral and lifetime changes.

Also for PPEB-2 (Figure 3 and lines 220 – 223) the spectral changes (shifts, changes in peak ratios) upon aggregation are relatively small, and only shown for a single example (is that representative? how does the histogram of 0-0/0-1 ratios for single chains and aggregates look like?). Moreover, the shorter lifetime of the PPEB-2 aggregate in this example is an exception (compare Fig. 4e, which clearly shows on average significantly longer lifetimes as compared to single chain!). Hence, assignment of the PPEB-2 aggregate emission to J-type behaviour does again not make much sense. The longer lifetimes in Fig. 4e rather suggest the formation of H-type structures with much weaker coupling strength as compared to PPEB-1; hence spectral changes are not as pronounced for PPEB-2. The slight red-shift from single chain to aggregate (Fig. 3a,c) is more likely to originate from the different dielectric environment (barely polarisable PMMA for single chains and the highly polarisable p-electron systems of PPEB-2 itself for aggregates) in combination with chain planarisation due to aggregation (which might be interpreted as J-type behaviour in terms of chemical repeating units but NOT in terms of “chromophoric units”).

6. Line 258: How can a dipole forbidden lowest excited state imply a red-shifted PL?

7. Lines 273 – 275 and Figure 4: If there is no change in photophysics for PPEB-3 between single molecule and aggregate, and if the absorption cross section in an aggregate is larger, why is the aggregate PL only a factor of about 2 larger? There are about 15 chains per aggregate, i.e. I would expect a PL increased by a factor of about 15. Or is there a further quenching mechanism at work? The antibunching data demonstrate about 15 emitters per aggregate.

8. Line 318: The authors claim that interchromophoric coupling is removed in a disordered aggregate. Then why is there inter-chain energy transfer in (disordered) conjugated polymer films (observed by many groups)?

9. In the antibunching section (lines 355 – 361) the authors state that “at least 17 chromophores within the aggregate behave as 1 – 2 chromophores in the H-aggregates ...”. This statement is wrong: The emitters in H-aggregates are not chromophores anymore, an H-aggregate is a collective system of many (strongly) coupled chromophores and behaves by definition as a single system. In this direction: What is meant by a single emissive centre in an H-aggregate (lines 397 – 398)?

10. Photon bunching in PPEB-2 aggregates was rationalised by the formation of CT states due to intra-chain J-type coupling (line 370). Yet, formation of such CT states in homopolymers requires H-type assembly, see refs. 42-45. Given that PPEB-2 aggregates are indeed H-type structures (and not J-type as claimed by the authors), this interpretation seems reasonable.

In summary, I cannot recommend publication of this manuscript in Nature Communications. Similar switching (H- vs. J-type) behaviour was already observed for single P3HT-nanostructures. Moreover, I find the data “overinterpreted”, i.e. rather than claiming J-type behaviour for the swollen PPEB-1 and dry PPEB-2 aggregates, I suggest to stick with aggregation induced planarisation of chains within the aggregates, which gives rise to the observed behaviour. Again, such behaviour is established and well-known for many conjugated oligomers and polymers (see Raithel et al., *Macromolecules* 2016, and Panzer et al., *J. Phys. Chem. Lett.* 2016).

Reviewer 1:

This is a fabulous paper, which demonstrates through carefully designed experiments the ability to tune between J and H-aggregates in conjugated polymers. Such polymers are technologically of great importance in a range of applications from light emitting diodes to solar cells.

The authors use a solvent swelling technique to modulate the packing distance between polymer chains within a given aggregate, effectively turning on and off the interchain coupling. This has a profound impact on the photo physical properties; in the emitting state of H-aggregates the chain dipoles are out of phase leading to a coherent suppression of the radiative decay rate and the 0-0/0-1 PL ratio. By contrast, isolated chains behave as J-aggregates where exciton coherence along the chain leads to enhanced radiative decay rates and 0-0/0-1 ratios. The experiments beautifully bear out these behaviors in quite dramatic fashion; the fluorescence decay rate and PL ratio is enhanced by an order of magnitude in J vs H-aggregates. The authors go on to further demonstrate H/J tuning via side-chain engineering. In addition, they show superior chain alignment within H-aggregates via polarization resolved single-aggregate PL measurements and utilize photon correlation techniques to determine the effective number of emitters. All of these experiments clearly reveal the intricate morphology-dependent photophysics at play in conjugated polymers. This paper will certainly appeal to a broad range of scientists interested in energy transport in organic materials and will no doubt generate considerable excitement especially with regard to the impact of coherence, which is also suspected of enhancing function in biological materials.

There is only one issue that needs to be addressed. On the first page the authors claim that the J-aggregate behavior of isolated chains arises from a head-to-tail arrangement of transition dipole moments a la Kasha theory. In fact, the main reason single polymer chains behave as J-aggregates is due to the strong covalent coupling between repeat units. Conjugated polymers

behave as direct band-gap semiconductors: the opposite curvatures of the conduction and valence bands gives rise to an exciton band which has positive curvature at $k=0$, which is the requirement for J-type photophysics. This is described in the paper, "Strong Photophysical Similarities between Conjugated Polymers and J-aggregates", *J. Phys. Chem. Lett.* 5, 622 2014. The positive curvature is directly linked to an effective coupling between neighboring repeat units which is negative(see Eq. 13) , as exists in J-aggregates. (There may be also contributions from the interacting transition dipoles which will enforce the J-like behavior of CPs if they are arranged in a head to tail fashion).

Hence, I would recommend that the authors slightly rephrase the introduction to reflect the comments above.

R1.0: This issue is also addressed by Reviewer 3 in points 2, 5 and we have rephrased the introduction and several parts of the manuscript to be more specific regarding the term J-type coupled units. We agree with both reviewers that it is always the covalent coupling between repeat units of the conjugated polymer chain, which leads to J-type PL characteristics of a single conjugated polymer chain. The degree of these J-type PL characteristics depends essentially on the number of repeat units and the strength with which they are coupled together. If more repeat units are coupled or if the coupling strength increases between repeat units, e.g. by planarization, which leads to improved pi-conjugation between repeat units, a stronger J-type PL will be observed, provided that static or dynamic (thermal) disorder does not impede coupling.

Since exciton coherence is an important theme in this paper, the authors may also want to mention (but this is not required) that for J-aggregates, like single-chain CPs, the PL ratio is a direct measure of the number of coherently coupled repeat units. Hence, the ratio grows with coherence length, as shown in *J. Chem. Phys.* 135, 054906 (2011) for CPs. The authors may want to reference some of Michel Schott's papers where he is able to grow effectively defect-free straight polydiacetylene chains which have a huge 0-0/0-1 ratio of about 100 (!) consistent with a very long coherence length of at least hundreds of nanometers, see Dubin, R. Melet, T. Barisien, R. Grousseau, L. Legrand, M. Schott, and V. Voliotist, *Nat. Phys.* 2(1), 32 (2006). Lecuiller, J. Berrehar, J. D. Ganiere, C. Lapersonne-Meyer, P. Lavallard, and M. Schott, *Phys Rev. B* 66(12), 125205 (2002)

R1.1: We thank the reviewer for pointing out the beautiful work by Michel Schott and we included the mentioned references in the discussion of increased coherent coupling between repeat units in the swollen PPEB-1 aggregates and dry PPEB-2 aggregates. We note, however, that even these polydiacetylenes are not perfect systems – the radiative lifetime is strongly temperature dependent due to thermal disorder, and despite the huge 0-0/0-1 ratio the radiative PL lifetime at low temperatures is very modest, approx. 300 ps, which is not that different from beta-phase polyfluorene where the ratio is much smaller. So something does not quite add up, but this is a topic for a future story.

Reviewer 3:

Eder et al. use single-molecule spectroscopy to study the conjugated polymer poly(p-phenylene-ethynylene) with different side groups that affect its aggregation behaviour. Controlled aggregation was achieved by solvent-vapour annealing. The aggregation, or more specifically, the electronic coupling between chains was investigated by PL spectroscopy,

lifetime measurements, excitation polarisation spectroscopy, and photon antibunching experiments on isolated aggregates / single chains. Extensive data sets were acquired comprising several hundreds of chains/aggregates.

R3.0: We thank the reviewer for the in-depth constructive feedback provided. We note that this short summary at the beginning of the review gives the impression that reviewer 3 was mainly focused on the second part of the manuscript in which the aggregation behaviour is controlled by different side groups of the polymer. We agree that this second part, viewed on its own, may appear related to previously published work (as reviewer 3 pointed out in the final statement), and we are happy to include the mentioned work, which is helpful to spell out a broader materials basis. But we do have to stress that this comparison to prior work misses a crucial point: all prior work on aggregation phenomena was carried out in the bulk film or concentrated solutions, where static and dynamic disorder masks many effects. The sub-ensemble based *mesoscopic* approach presented here yields unprecedentedly clear connections between the morphology, the predominant type of electronic coupling, the energy transfer properties and the nature of dark-state formation. Further, the manuscript has to be seen as a whole including the first part in which we actually demonstrate, for the first time, the *reversible switching* between H-type and J-type PL from single mesoscopic objects. Such clear switching on one and the same single object as observed in the PL spectrum and the PL lifetime simultaneously has not been previously reported (see discussion under R3.3), and provides unique access to the underlying physics of aggregation.

We initially planned to submit the manuscript based solely on the first part, but then decided to include the second part to illustrate the broad applicability of the concept and the conceptual connection to prior work. If the referee feels very strongly about the novelty of the second part, we would be willing to place it in the Supplementary Information, although we stress again that the approach using single aggregated objects is, strictly, entirely novel.

However, there are several issues, in particular with the interpretation of data that prevent publication in present form:

1. The expression “coherent coupling mode” is very unusual (I never came across that so far). Either H-type or J-type coupling is meant. In both cases the interaction is the same (Coulomb-coupling), there is no fundamentally different physics involved. Please stick to the common notation in the field.

R3.1: By the word “mode” we meant H-type or J-type coupling and we are switching between these two modes. To be more precise we changed the title to: “Switching between H- and J-type electronic coupling in single conjugated-polymer aggregates”. We also changed “coherent coupling mode” to “type of electronic coupling” throughout the manuscript. We hope that the referee agrees that this is really semantics. The level splitting is a result of coherent delocalization of excitation energy by the Coulomb interaction, but it is probably not necessary to reiterate this at every corner.

2. Related to point 1 above: The authors refer to intra-chain (J-type) and inter-chain (H-type) coupling between “neighbouring chromophoric units” (l. 39); and “J-type coupling occurs predominantly between chromophoric units on the same CP chain ...” (l. 42/43). Here I miss clear definitions of “chromophoric units”. In this manuscript, intra-chain J-type coupling

appears to me to be the interaction between chemical repeating units of a chain, and NOT interactions between chromophores of a chain (if I do understand the sketches in Fig. 1a correctly). It is worth reading Spano's paper on HJ-aggregates carefully (J. Chem. Phys. 2013). He defines precisely what is interacting with what on a chain and between different chains. The introduction and several parts of the manuscript have to be clarified / rewritten in this respect.

R3.2: The same point was raised by reviewer 1 and we rephrased the introduction and several parts of the manuscript to clarify the definition of H-type and J-type couplings. In this context we also included Yamagata *et al.* J. Chem. Phys. 139, 114903 (2013) in the discussion. See also discussion in R1.0.

3. In lines 92-94 the authors claim that switching between H- and J-type coupling was only predicted so far (which again requires a clear definition of those interactions). However, such behaviour was already observed for P3HT-Nanofibres by Grey *et al.* (J. Phys. Chem. B 2012, J. Phys. Chem. Lett. 2012), and to some extent for MEH-PPV by A. Köhler *et al.* (J. Chem. Phys. 2013).

R3.3: Lines 92-94 originally read: “*This observation confirms the predicted simultaneous presence of H- and J-type coherent coupling modes in CPs, and demonstrates that H-type coupling can completely mask the J-type coupling.*” First, we are generally careful in using the expression “so far” and usually abstain from using it (as we did here). Second, the work by Grey *et al.* and Köhler *et al.* demonstrates that the degree of inter-chain coupling, i.e. H-type coupling, can be altered by applying pressure or decreasing the temperature. Here, we demonstrate that the interchain coupling can be *completely* switched off reversibly by partially swelling the aggregates. Surprisingly, we observe that the remaining PL of the aggregated chains shows stronger J-type character compared to single chains, a crucial result not demonstrated previously. To stress this central aspect of our work, we extracted the full width at half maximum (FWHM) of spectra of swollen PPEB-1 aggregates and dry single PPEB-1 chain spectra. All spectra are shown in the Supporting Information in Figure S7, and the FWHM values are plotted in a new histogram in Figure 2f. This complete switching between H-type and J-type coupling confirms that aggregation leads to stronger intrachain J-type character as compared to single chains, an effect which is completely masked by the interchain H-type coupling in the non-swollen aggregates.

To be more precise, we changed the text (previous lines 92-94) to: “*This observation confirms the predicted simultaneous presence of H- and J-type electronic coupling types in CPs, and demonstrates that interchain H-type coupling can completely mask the increased intrachain J-type coupling arising from improved chain ordering in the aggregates.*”

4. The “dry” PL spectra of PPEB-1 in Fig. 2d possess a very strange shape, which is not consistent with the emission of a single H-type aggregate although the highest energy peak is indeed suppressed; yet, the 0-1/0-2 ratio does not fit. This spectrum rather appears to be a superposition of several H-aggregates emitting at different spectral positions, consistent with on average 1-2 emitters shown by antibunching experiments.

R3.4: The point here is that a perfect H-aggregate does not emit by itself. Radiation comes from an excimer-like transition (see e.g. Stangl *et al.*, J. Phys. Chem. Lett. 6, 1321 (2015)), but since the aggregate also exists in the ground state, there is some residual vibronic structure in the excimer band. Because the (dipole-forbidden) transition lifetime varies so much from aggregate to aggregate, there is also a large variability in the PL spectrum of the H-aggregate which, of

course, is masked in ensemble measurements. There is nothing “strange” about this, this is a feature of subensemble spectroscopy. Because of the excimeric nature of the emission, which strongly broadens and shifts the spectrum, the 0-1/0-2 ratio in comparison to the monomer emission is not the easiest parameter to look out for. In addition, as demonstrated in Walter *et al.* J. Am. Chem. Soc. 130, 16830 (2008), C=C and C≡C modes can mix in the vibronic progression of the PL, showing up as an additional band in PL (see below).

Given the thermally induced inhomogeneous broadening of the aggregate PL, it is not meaningful to extract vibronic peak ratios in the PL.

5. Line 194: “Narrow 0-0 peaks” from “J-type coupled chromophores” are mentioned. What is narrow? Narrow compared to what? Please quantify. Moreover, the “swollen” PPEB-1 aggregate spectrum looks very similar to that of a single PPEB-1 chain, the lifetimes are essentially identical (cf. Fig. 2d,e and 3a,b). On which basis is the “swollen” spectrum then assigned to that of a J-aggregate?

R3.5: We apologize for not being more specific at this point. We now analysed hundreds of spectra of the single PPEB-1 chains and the swollen PPEB-1 aggregates to compare the FWHM of the spectra (see Figure S7). The histograms of these FWHM values are shown in Figure 2f and demonstrate that the FWHM of the swollen PPEB-1 aggregates is indeed smaller compared to single PPEB-1 chains. The width of the 0-0 peak is directly connected to the number of coherently coupled dipoles on the polymer chain, due to exchange narrowing (see e.g. Walczak *et al.*, J. Chem. Phys. 128, 044505 (2008)). We conclude that the swollen PPEB-1 aggregates exhibit more J-type character as compared to single PPEB-1 chains, owing to a higher degree of intrachain ordering, which is surely quite a surprising result.

The “swollen” chains might be more planar than a single isolated chain (e.g. P3HT planarises before it forms H-aggregates). But why is that potentially more planar chain then a J-aggregate (Fig. 2d), for which the authors claim that several “chromophoric units” along the same chain

interact? Why is the single chain then not a J-aggregate (Fig. 3a)? This discrimination makes no sense based on such very minor spectral and lifetime changes.

See discussion of R1.0 and R3.2. We have reworked the manuscript regarding the definition of J-type coupled aggregates and we abstain from using the term “chromophoric units” in this context to prevent confusion. We agree with the reviewer that the swollen aggregates exhibit a *higher degree* of intrachain order as compared to single chains, which might be due to planarization of the repeat units. Of course, a single polymer chain can be viewed as a J-type coupled 1D aggregate in which the covalently bound repeat units couple together electronically (see reviewer 1). The additional plot in Figure 2f (see also R3.3) now quantifies the spectral differences between single chains and swollen aggregates in more detail: swollen aggregates have a FWHM of the PL spectra of 80 ± 1 meV compared to a FWHM of 100 ± 1 meV for the single chains. We note that the errors regarding the mean FWHM value are the standard error of the mean and not the standard deviation of the distribution, which is larger. We therefore conclude that the swollen aggregates demonstrate a *higher degree* of J-type character as compared to single chains. According to Knapp in Chem. Phys. 85, 73-82 (1984), this difference in linewidth corresponds to an increase of ~ 1.6 times in the number of dipoles coupled within the chains in the swollen aggregates compared to single chains.

Also for PPEB-2 (Figure 3 and lines 220 – 223) the spectral changes (shifts, changes in peak ratios) upon aggregation are relatively small, and only shown for a single example (is that representative? how does the histogram of 0-0/0-1 ratios for single chains and aggregates look like?).

We have measured hundreds of additional spectra from single chains and aggregates of PPEB-1, PPEB-2 and PPEB-3, which are shown in the new Figure 3, and extracted the FWHM of all spectra. Figure 4 (previously Figure 3) now includes a histogram of the measured FWHM values for all samples with an additional discussion in the manuscript. The FWHM values of the dry PPEB-2 aggregates are even lower as compared to the swollen PPEB-1 aggregates, with on average 70 ± 1 meV, demonstrating that these dry PPEB-2 aggregates exhibit the strongest intrachain J-type coupling. The spectral narrowing corresponds to an increase of ~ 2 times in the number of dipoles coupled compared to single chains following Knapp’s model. The analysis of the spectral FWHM is a much clearer measure of the degree of J-type coupling than an analysis of the 0-0/0-1 peak ratio, since absolutely no fitting is required and the values are simply extracted from the raw data. As noted above, different vibrational modes can overlap or even mix, so a clear assignment of spectral features is not always possible.

Moreover, the shorter lifetime of the PPEB-2 aggregate in this example is an exception (compare Fig. 4e, which clearly shows on average significantly longer lifetimes as compared to single chain!). Hence, assignment of the PPEB-2 aggregate emission to J-type behaviour does again not make much sense. The longer lifetimes in Fig. 4e rather suggest the formation of H-type structures with much weaker coupling strength as compared to PPEB-1; hence spectral changes are not as pronounced for PPEB-2.

This point was already discussed in the original manuscript: “*However, a similar correlation between F_{red} and τ_{PL} to that found in PPEB-1 aggregates is seen in Figure 4e, indicating that H-type coupling still occurs but is less dominant. However, H-type coupling does not swamp the intrachain J-type coupling in this case, as is evidenced by the increased average PL intensity with respect to PPEB-1 aggregates.*”

However, the new data set provided in the new Figure 3 shows the decreased FWHM of these aggregates, which clearly demonstrates the more pronounced intrachain J-type coupling in PPEB-2 aggregates compared to single chains. This observation does not exclude the possibility of an additional interchain H-type coupling component for a sub-population of aggregates, which would affect the PL lifetime and the F_{red} value. As we demonstrated in Figure 2, both coupling types can be present at the same time. It is important to note that the data set in (now) Figure 5 is recorded differently as compared to the new Figure 3. The F_{red} value, τ_{PL} and the PL intensity in Figure 5 are recorded by obtaining confocal scan images and subsequently evaluating each diffraction-limited spot with a spot-recognition software. Therefore, each particle was only illuminated for approximately 100-200 ms. In contrast, the spectra in Figure 3 were obtained by 4 s integration of the PL. The different acquisition methods can lead to subtle differences due to photoselection and photodegradation of the emitter, e.g. H-type emitters within the aggregate can bleach with time leaving behind J-type emitters. For this reason, a larger sub-population of H-type emitters is observed in Figure 5 as compared to Figure 3. A small paragraph was inserted to discuss these differences, which are a necessary consequence of the sub-ensemble approach.

The slight red-shift from single chain to aggregate (Fig. 3a,c) is more likely to originate from the different dielectric environment (barely polarisable PMMA for single chains and the highly polarisable p-electron systems of PPEB-2 itself for aggregates) in combination with chain planarisation due to aggregation (which might be interpreted as J-type behaviour in terms of chemical repeating units but NOT in terms of “chromophoric units”).

We agree that especially the line narrowing is a strong indication of increased J-type behaviour, quiet possibly due to planarization of the chain (see R1.0, R3.2 and R3.5). We state this now explicitly in the manuscript.

6. Line 258: How can a dipole forbidden lowest excited state imply a red-shifted PL?

R3.6: A dipole “forbidden” lowest excited state does not imply that no PL can be emitted from this state (as in the case of, e.g., phosphorescence). What this really means is that the radiative rate is significantly reduced. But provided that non-radiative decay is negligible, the PL intensity will not be affected significantly. This situation can only be reached on the single-molecule level, as there is always some quenching present in the ensemble. We previously discussed this often observed fact in detail using a model system for H-type coupled chromophores (Stangl *et al.*, J. Phys. Chem. Lett. 6, 1321 (2015)). We have reiterated this crucial point in the manuscript. The PL is excimeric in nature. The 0-0 transition can, in principle, still be observed in H-type coupled aggregates, but this is red-shifted compared to the monomer emission since the energy levels in the excited state are split. However, the most significant contribution to the red-shift in PL stems from the decreased 0-0/0-1 peak ratio (see red spectrum in Figure 2d). For this reason, the overall integrated PL is red shifted for an H-type aggregate (in which the lowest excited state is dipole forbidden) compared to single chains or aggregates without significant H-type coupling. We inserted a short explanation in the manuscript to avoid confusion on this point. Also, we changed the sentence in line 258 to “*The transition from the lowest excited state to the ground state becomes dipole forbidden, hence the increase of τ_{PL} and the red-shifted PL.*” to be more precise.

7. Lines 273 – 275 and Figure 4: If there is no change in photophysics for PPEB-3 between single molecule and aggregate, and if the absorption cross section in an aggregate is larger, why

is the aggregate PL only a factor of about 2 larger? There are about 15 chains per aggregate, i.e. I would expect a PL increased by a factor of about 15. Or is there a further quenching mechanism at work? The antibunching data demonstrate about 15 emitters per aggregate.

R3.7: We did make the assumption that the physical aggregation process of PPEB-3 during SVA is similar to that of the other compounds: under SVA the behaviour in terms of the decrease of spot density and the brightening of individual fluorescence spots really is pretty much identical. But of course, this assumption might not be entirely correct. Due to the bulky side-chains of PPEB-3 the solubility is increased significantly. Therefore, the saturation concentration during solvent vapor annealing will be increased, which leads to an increased critical radius for stable aggregate formation during Ostwald ripening and an increased number of “left-over” single chains (see Vogelsang *et al.* Nature Mater. 10, 942 (2011)). The average aggregate size is still determined as 15 chains per aggregate according to the number of diffraction-limited spots seen in the microscope before and after SVA, but the distribution of aggregate sizes may be more heterogeneous for PPEB-3 as compared to the other samples. The intensity distribution in Figure 4 most likely reflects this strong heterogeneity, and the bias to lower PL intensities probably stems from an increased number of “left-over” single chains. However, this bias actually indicates that even smaller PPEB-3 aggregates (compared to the other two samples) – or even single PPEB-3 chains – actually display *more* independent emitters, based on their photon statistics, than found in the other two materials (where the objects studied are most likely larger). This counterintuitive observation is now mentioned in the manuscript, but the main conclusions remain the same: electronic coupling (H-aggregation) is never observed in PPEB-3 aggregates and the interchromophoric excitation energy transfer properties, as determined by the photon statistics, are much less pronounced.

8. Line 318: The authors claim that interchromophoric coupling is removed in a disordered aggregate. Then why is there inter-chain energy transfer in (disordered) conjugated polymer films (observed by many groups)?

R3.8: It is well known that small changes to side chains can have a dramatic impact on interchromophoric energy transfer, see, e.g., Müllen’s dendronized polyfluorene or Burn’s small-molecule dendrimers (Lupton *et al.* Phys. Rev. B, 66, 155206 (2002)). But the effect of spacing side chains is, of course, much greater on coherent coupling (i.e. H-aggregation) than on incoherent interchain energy transfer, which the reviewer refers to here. But there is an additional issue. The reviewer compares here a single disordered aggregate with a disordered conjugated polymer film, which is problematic. Energy transfer properties in conjugated polymer films are measured by averaging over a large area, at least a diffraction limited area, containing billions of molecules and molecular conformations. Within this area different degrees of disorder are to be expected and it is not known how interchain energy transfer is weighted and if the measured interchain energy transfer reflects the entire disorder in the system. In other words, a polymer film, which has the same morphology throughout the entire film as our disordered PPEB-3 aggregates, might indeed have poor interchain energy transfer properties. However, it is not under debate that interchain energy transfer is hampered in disordered systems compared to ordered systems (see e.g. Hedley *et al.* Chem. Rev. 117, 796 (2017)) and we mention this know briefly in the manuscript.

9. In the antibunching section (lines 355 – 361) the authors state that “at least 17 chromophores within the aggregate behave as 1 – 2 chromophores in the H-aggregates ...”. This statement is wrong: The emitters in H-aggregates are not chromophores anymore, an H-aggregate is a

collective system of many (strongly) coupled chromophores and behaves by definition as a single system. In this direction: What is meant by a single emissive centre in an H-aggregate (lines 397 – 398)?

R3.9: This is a bit of a semantic issue – the term chromophore has a well-defined meaning in the context of a conjugated polymer, but in terms of an optical entity one could also refer to H-aggregated multiple chromophores as a single chromophore since the quantum-optical definition of a chromophore is that of a single emitter. This is, of course, a bit confusing and should really be avoided in the paper. We agree with the reviewer that an H-aggregate is a collective system of many coupled chromophores and we did somewhat over-simplify the discussion here, giving rise to some confusion. However, we would also like to stress that we only stated in the manuscript “*behave as 1-2 chromophores*”, which does not imply that we actually “*have*” 1-2 chromophores. Therefore, this statement is not wrong, it is a simple observation based on the quantum-optical characteristics. To clarify further, a chromophore is seen as a single emissive unit, which can only emit one single photon at a time in the context of photon statistics. This single emissive unit can be a single chromophore, but also a collective system of many coupled chromophores, which *behave* with respect to their photon statistics as a single chromophore. In this context, what we mean by a single emissive centre in an H-type coupled aggregate is exactly such a collective system of many coupled chromophores and we cannot rule out that there are multiple such emissive centres within an H-type coupled aggregate, because the antibunching is not perfect. To avoid confusion, we changed the term “chromophore” here to “emissive unit” or “single photon emitter” in the antibunching section and provide a clear definition of such an “emissive unit” based on its photon statistics: “*Such a single emissive unit can only emit one single photon during its excited state lifetime and may consist of either a single chromophore or multiple chromophores coupled together by coherent or incoherent energy transfer*”.

10. Photon bunching in PPEB-2 aggregates was rationalised by the formation of CT states due to intra-chain J-type coupling (line 370). Yet, formation of such CT states in homopolymers requires H-type assembly, see refs. 42-45. Given that PPEB-2 aggregates are indeed H-type structures (and not J-type as claimed by the authors), this interpretation seems reasonable.

R3.10: As outlined above, the interchain H-type coupling is suppressed in PPEB-2 aggregates so that intrachain J-type coupling (the improved ordering of the packed chains compared to single chains) becomes apparent. The PL spectra of PPEB-2 aggregates are narrower than those of single PPEB-2 chains: how else would one explain this if not by intrachain J-aggregation? The referee is, of course, correct that chromophore ordering required for CT state formation and H-aggregation are basically equivalent, but the interaction lengths for the two processes are different so that (short-range) H-aggregation can be turned off while still preserving the necessary chromophoric proximity for CT state formation. However, we agree that the assignment of the photon bunching to CT states versus triplet excitons may not be quite as unambiguous as we originally made out and so we have reworked the paragraph accordingly.

The measurements were carried out under ambient conditions (air), which usually leads to efficient quenching of triplet states. It therefore seems reasonable to claim that all triplets are quenched and the dark-state observed in the photon bunching is therefore assigned to CT-state formation. However, it is also true that the exposure to oxygen of the aggregates embedded in the PMMA film may be limited if they are located deep within the inert matrix. As we recently demonstrated, even a few tens of nanometers can have a dramatic effect on the triplet lifetime,

effectively shielding the triplet from the oxygen (see Würsch *et al.* J. Phys. Chem. Lett. 7, 4451 (2016)). So just claiming the presence of oxygen in the measurement is not sufficient to conclude that the bunching arises from CT-state formation.

It is very interesting to note that comparison with literature also suggests that triplet states are more easily formed in J-type aggregates than in isolated chains (Thomas *et al.* ACS Nano, 8, 10559 (2014), Thomas *et al.* J. Phys. Chem. C 120, 23230 (2016)), a previous claim which appears to be consistent with our observations here.

The latter reference proposed that “*the high intrachain order in purified aggregates that extends exciton coherence lengths, leading to J-aggregate spectral signatures, is also important for populating interchain charge transfer (CT) states that, at longer times, recombine preferentially to triplets according to spin statistics*”. We toned down our interpretation regarding these dark-states and now note that the dark states responsible for photon bunching may be triplets, which preferentially arise in J-type coupled aggregates. We also note that this is an ongoing question of research and point out our recent discovery that triplets in P3HT are actually quenched (presumably by CT state formation) in multi-chain aggregates (Steiner *et al.* J. Am. Chem. Soc. 139, 9787 (2017)).

In summary, I cannot recommend publication of this manuscript in Nature Communications. Similar switching (H- vs. J-type) behaviour was already observed for single P3HT-nanostructures. Moreover, I find the data “overinterpreted”, i.e. rather than claiming J-type behaviour for the swollen PPEB-1 and dry PPEB-2 aggregates, I suggest to stick with aggregation induced planarisation of chains within the aggregates, which gives rise to the observed behaviour. Again, such behaviour is established and well-known for many conjugated oligomers and polymers (see Raithel *et al.*, Macromolecules 2016, and Panzer *et al.*, J. Phys. Chem. Lett. 2016).

We reiterate our gratitude for the helpful comments provided by the reviewer but repeat here that we probably simply offered too much material in a single paper. There is only one prior report in the literature (ours) of single deterministic multi-chain polymer aggregates, and nobody has ever demonstrated the reversible switching within one and the same single aggregate between J- and H-type coupling using SVA.

Only on the level of single aggregates can one resolve the residual enhanced J-type aggregation once H-type coupling is attenuated by solvent swelling. This surely is a near-perfect demonstration of the subtle interplay between the two coupling modes depending, very sensitively, on interchain spacing. This result should be much more obvious to the reader now by the additional discussion of the statistics of PL spectra and PL linewidth, which we originally – regrettably – omitted from the manuscript.

Again, there appear to be mainly semantic objections at play here, and we are keen to work with the reviewer to resolve these. By any definition, planarization of the polymer chain induced by aggregation constitutes improved intra-chain J-type coupling. The only issue is that the single-molecule approach has no immediate observable for planarization (unlike, e.g., structural analysis by x-ray scattering in films): all we can say is that the intrachain coherence improves as witnessed by the red shift and the spectral narrowing compared to isolated chains.

Since the single-molecule approach is unique and not comparable in all aspects to ensemble studies, we tried to limit the comparison of our original results to the prior literature. However, we are happy to include the additional two references mentioned and stress here that we are

really probing direct observables of the coherent coupling – the spectrum, lifetime, intensity, polarization, photon statistics, etc. – of a single entity which can only exhibit one specific type of coupling, rather than probing ensembles where disorder broadening dominates the observables.

Reviewers' Comments:

Reviewer #2:

Remarks to the Author:

Overall, the authors responded well to the Reviewer criticisms and the manuscript is improved. The paper presents a unprecedented view of the interplay between J- and H-aggregate mechanisms in CP aggregates. The dramatic difference in the spectral properties and photophysics in general between the H (dried) and J (swelled) forms and the controllable switching between the two disparate forms/behaviors is quite novel. The paper fully deserves publication in Nature Communications after a few simple changes are made,

1) the segment on line 38, "and is built upon... coupling according to Kasha's exciton theory" is a bit confusing. I suggest the following rewording:

"and is built upon a combination of J- and H-type couplings; unconventional J-type coupling between covalently coupled repeat units within a polymer chain and conventional H-type coupling between cofacial chromophoric units on neighboring chains which abides by Kasha's exciton theory.""

2) On line 44 "because a head-to-tail arrangement of the interacting transition dipole moments is necessary". This is true in conventional van der Waal aggregates where the positive band curvature defining J-aggregates is caused by the head-to-tail Coulomb coupling (a la Kasha), but it is NOT true for polymer chains where the positive band curvature (and unconventional J-aggregate behavior) arises primarily from the covalent interactions between repeat units. The authors should either omit the statement or change it to reflect the above.

(In fact, I believe it may be a simple oversight, because the authors later explain that the origin of the "positive band curvature" (line 53) results from covalent interactions.)

3) Line 486; "prising" should be "prying"

4) Line 489; The segment beginning "Second, by fine tuning..." is not a sentence. Please correct.

Reviewer #3:

Remarks to the Author:

In general, I am happy with the revision provided by Eder et al. In particular, the additional spectra and line width data make this work more complete and more convincing in the interpretation of the different H- and J-type couplings in the different materials/aggregates investigated. This manuscript is publishable given the following minor issues are addressed:

1. Around lines 230 and 290 the authors refer to Knapp's work to determine changes in delocalisation of electronic excitations in aggregates from changes in spectral line widths. However, Knapp assumes in his analysis that correlations of transition energies of the building blocks (here repeat units) of an aggregate (here along a single CP chain) are entirely absent. Since this critical prerequisite is very likely not completely fulfilled in conjugated polymers (see Collini, Scholes, Science 2009), I strongly suggest that the authors do not provide hard numbers for the change in delocalisation based on this model. This discussion should rather be kept on a qualitative level.

2. In the description/discussion of Fig. 4d-f, PPEB-3 is never mentioned.

As a final comment: Some of my issues were indeed semantic in nature. But e.g. using the expression "chromophore" or "chromophoric unit" with three different meanings in a single manuscript without providing clear definitions is not "a bit of a semantic issue" (R.3.9). Particularly in a large manuscript with extensive data sets and several techniques employed, this does not add to clarity and confuses readers. Also, the authors could have used established language in first place already, without trying to rephrase known phenomena with new expressions (What is a "coherent coupling mode"? Does that sound more novel?). This is annoying and not helpful.

Reviewer #4:

Remarks to the Author:

This is an interesting paper, which demonstrates the ability to find-tune between J and H aggregates in different conjugated polymers. Through a large set of experiments, the authors further demonstrated the importance of side-chain modification on the formation of different aggregates for conjugated polymers. The authors also took the revision process quite seriously and a large amount of work has been done during revision, and the manuscript has also been improved based on the reviewers' comments. The study is of fundamental importance to our understanding of functional materials, with the hope that it can further guide our future design of conjugated polymers in a way that is more precise. The paper will certainly appeal to the wide audience in optical materials, energy materials and even optoelectronic devices.

The manuscript is suitable for publication after addressing the following issues:

- 1) Although the authors have agreed to change coherent coupling in the revised manuscript, but it still appears in the abstract.
- 2) There has been argument about the repeat units and chromophore center, especially in H aggregates. I'm not sure how the chromophore is defined in a polymer chain.
- 3) Is there any change in the torsional angle between the swollen chains, H or J aggregates? Should that also be reflected in the emission spectra?
- 4) As mentioned by the reviewer 3, the switching between J and H aggregates was reported for P3HT before, can the authors make a comment and offer justification to highlight the difference in the contribution between the papers?
- 5) The authors mentioned that the H-type emitters within the PPEb-2 aggregates may bleach with time, which leaves behind J-type of emitter. What is the reason for J-type to be more photo-resistance?
- 6) Several lines above the Methods, the author mentioned that the dark states are quenched by interchromophore (not really sure what it refers to) interactions in the H-aggregates, which is in analogy to the strong quenching of triplets in P3HT. I'm not sure whether in the PPE system, the quenching is also associate with triplet or not.

Dear Editor,

We are delighted to see that Reviewer 3 respects the substantial additional effort we invested in the resubmission and now shares our enthusiasm for the work along with the other two reviewers; indeed, the new Rev. 4 seemed quite impressed by our response to Rev. 3. We thank all reviewers for their thorough review of our work and the constructive suggestions for improvement. We have responded to all points raised in detail and modified the manuscript appropriately.

The responses are labelled by reviewer and comment number, e.g. R1.1 for reviewer 1 and comment 1.

Reviewer 2:

Overall, the authors responded well to the Reviewer criticisms and the manuscript is improved. The paper presents a unprecedented view of the interplay between J- and H-aggregate mechanisms in CP aggregates. The dramatic difference in the spectral properties and photophysics in general between the H (dried) and J (swelled) forms and the controllable switching between the two disparate forms/behaviors is quite novel. The paper fully deserves publication in Nature Communications after a few simple changes are made,

1) the segment on line 38, "and is built upon... coupling according to Kasha's exciton theory" is a bit confusing. I suggest the following rewording:

"and is built upon a combination of J- and H-type couplings; unconventional J-type coupling between covalently coupled repeat units within a polymer chain and conventional H-type coupling between cofacial chromophoric units on neighboring chains which abides by Kasha's exciton theory."

R2.1: Thank you. We changed the sentence accordingly.

2) On line 44 "because a head-to-tail arrangement of the interacting transition dipole moments is necessary". This is true in conventional van der Waal aggregates where the positive band curvature defining J-aggregates is caused by the head-to-tail Coulomb coupling (a la Kasha), but it is NOT true for polymer chains where the positive band curvature (and unconventional J-aggregate behavior) arises primarily from the covalent interactions between repeat units. The authors should either omit the statement or change it to reflect the above.

(In fact, I believe it may be a simple oversight, because the authors later explain that the origin of the "positive band curvature" (line 53) results from covalent interactions.)

R2.2: We changed the sentence to: "*J-type coupling occurs predominantly along the same CP chain, because of the covalent interactions between the head-to-tail arranged transition dipole moments (TDMs) of the repeat units^{13,15}.*"

3) Line 486; "prising" should be "prying"

R2.3: Apologies. We corrected the typo.

4) Line 489; The segment beginning "Second, by fine tuning..." is not a sentence. Please correct.

R2.4: We changed the sentence to: *“Second, by fine tuning the SVA aggregation process, isolated aggregates of different samples but of comparable sizes can be grown.”*

Reviewer 3:

In general, I am happy with the revision provided by Eder et al. In particular, the additional spectra and line width data make this work more complete and more convincing in the interpretation of the different H- and J-type couplings in the different materials/aggregates investigated. This manuscript is publishable given the following minor issues are addressed:

1. Around lines 230 and 290 the authors refer to Knapp's work to determine changes in delocalisation of electronic excitations in aggregates from changes in spectral line widths. However, Knapp assumes in his analysis that correlations of transition energies of the building blocks (here repeat units) of an aggregate (here along a single CP chain) are entirely absent. Since this critical prerequisite is very likely not completely fulfilled in conjugated polymers (see Collini, Scholes, Science 2009), I strongly suggest that the authors do not provide hard numbers for the change in delocalisation based on this model. This discussion should rather be kept on a qualitative level.

R3.1: The reviewer raises a good point here and we now abstain from providing definitive numbers on the increase of the number of coupled repeat units. We changed the following sentences to a more qualitative statement: *“According to the model by Knapp, this decrease in linewidth corresponds to an increase in the number of repeat units coupled in the chains in the swollen aggregates compared to isolated chains³⁹.”*

and

“By invoking the model of Knapp again, this difference in linewidth corresponds to an increase in the number of repeat units coupled along a single chain in the aggregate compared to the isolated single chain³⁹.”

2. In the description/discussion of Fig. 4d-f, PPEB-3 is never mentioned.

R3.2: We apologize for this omission and inserted a sentence: *“In contrast, comparison of the PPEB-3 single chain (Figure 4a-c, green data) and aggregate (Figure 4d-f, green data) results demonstrates that the bulky side chains of PPEB-3 prevent any kind of electronic coupling as is evidenced by the absence of any differences between these data sets.”*

As a final comment: Some of my issues were indeed semantic in nature. But e.g. using the expression “chromophore” or “chromophoric unit” with three different meanings in a single manuscript without providing clear definitions is not “a bit of a semantic issue” (R.3.9). Particularly in a large manuscript with extensive data sets and several techniques employed, this does not add to clarity and confuses readers. Also, the authors could have used established language in first place already, without trying to rephrase known phenomena with new expressions (What is a “coherent coupling mode”? Does that sound more novel?). This is annoying and not helpful.

R3.3: We apologize again for the confusion by apparently not providing sufficiently clear definitions. We completely agree with the reviewer that clear definitions and the use of established language are essential for the understanding, but we do note that there are substantial differences in terminology between different fields when referring to the same thing. We have removed the term “*coherent coupling mode*” everywhere and now say “*the form of coherent coupling, i.e. J- or H-type*” – surely this is now watertight. Naturally, we thank the reviewer for the patience brought forward in improving our presentation.

Reviewer 4:

This is an interesting paper, which demonstrates the ability to find-tune between J and H aggregates in different conjugated polymers. Through a large set of experiments, the authors further demonstrated the importance of side-chain modification on the formation of different aggregates for conjugated polymers. The authors also took the revision process quite seriously and a large amount of work has been done during revision, and the manuscript has also been improved based on the reviewers' comments. The study is of fundamental importance to our understanding of functional materials, with the hope that it can further guide our future design of conjugated polymers in a way that is more precise. The paper will certainly appeal to the wide audience in optical materials, energy materials and even optoelectronic devices. The manuscript is suitable for publication after addressing the following issues:

1) Although the authors have agreed to change coherent coupling in the revised manuscript, but it still appears in the abstract.

R4.1: We changed “coherent” to “electronic” in the abstract. We still note, however, that the actual process is coherent – if it were not coherent there would be no energy shift. Nevertheless, we do tend to side with the reviewer that the term “coherent” is simply overused in literature, even if it is not necessarily overstretched.

2) There has been argument about the repeat units and chromophore center, especially in H aggregates. I'm not sure how the chromophore is defined in a polymer chain.

R4.2: It is fairly straightforward to identify what a chromophore is in a conjugated polymer by comparing the spectral properties of oligomers to those of polymers. This approach necessitates a minimization of inhomogeneous broadening, which arises both from static and dynamic disorder, as well as thermal spectral broadening. At low temperatures, universal signatures of single chromophores can be identified in the fluorescence spectra (see Schindler *et al.*, Proc. Natl. Acad. Sci. U.S.A. 101, 14695 (2004) and Angew. Chem. Int. Ed. 44, 1520 (2005)). At room temperature, as in our present study, thermal broadening of the PL spectrum can become comparable to interchromophoric disorder, so that a PL spectrum will not necessarily yield the spectral information on a single chromophore. But this does not mean that the concept of a chromophore in the CP is poorly defined.

What is indeed interesting is that the excimer-like emission feature of the H-aggregate can indeed be thought of as a single "chromophore", not least because it can show photon antibunching. But actually, of course, it consists of multiple CP chromophores. This fact is indeed a bit confusing, so we have added a sentence to the manuscript to stress this point:

"While the concept of a chromophore on a CP chain is, in principle, well defined as a finite number of repeat units¹⁶, which give rise to universal spectral features at cryogenic

temperatures^{30, 64}, the term "chromophore" really implies the region of the material, which lends it a particular colour. As such, the excimer-like emission of the H-aggregate can also be attributed to a "chromophore", showing photon antibunching. In this case, however, the emitting region actually consists of multiple CP chromophores, which couple together coherently to form one emitting centre."

3) Is there any change in the torsional angle between the swollen chains, H or J aggregates? Should that also be reflected in the emission spectra?

R4.3: The reviewer is of course right to speculate on the possibility of a change in torsional angle. In polyfluorene, for example, this is a well-known effect of SVA, which has dramatic consequences on cryogenic single-chain PL spectra (see Becker *et al.*, J. Am. Chem. Soc. 127, 7306 (2005)). To make a conclusive statement on this, however, we would need cryogenic PL spectra of the aggregates. The problem here is, unfortunately, that the SVA technique does not lend itself readily to cryogenic studies, where the solvent would freeze. In addition, both triplet formation and the nature of H-aggregate emission complicate low-temperature measurements (since the excimer lifetime increases with decreasing temperature, lowering the photon emission rate), so we have not succeeded in completing these measurements yet.

An alternative approach to probe torsion would be to measure Raman spectra, but this is tricky on single aggregates at room temperature because of the strong Rayleigh scattering. Also, unlike PL, the technique cannot discriminate between different regions of the aggregate (i.e. more or less ordered regions), so it is not clear that the results would be that unambiguous. In polyfluorene, for example, PL is a much clearer reporter on torsional angles than Raman.

We have added the following to the manuscript:

"The torsional angles between the monomers may change upon SVA and aggregation. A well-known example of this effect is found in polyfluorene, which can transition from the twisted glassy phase to the planarized beta-phase under SVA and gives rise to a dramatic change in vibrational modes in cryogenic single-chain PL spectra⁴⁶. We expect that a similar effect will arise in PPEB aggregates, but testing this will require combining the SVA technique with cryogenic SMS."

4) As mentioned by the reviewer 3, the switching between J and H aggregates was reported for P3HT before, can the authors make a comment and offer justification to highlight the difference in the contribution between the papers?

R4.4: As mentioned in the response to reviewer 3, the work by Grey *et al.* and Köhler *et al.* demonstrates that the degree of inter-chain coupling, i.e. H-type coupling, can be altered by applying pressure or decreasing the temperature. Here, we demonstrate that the interchain coupling can be *completely* switched off – reversibly – by partially swelling the aggregates. Surprisingly, we observe that the remaining PL of the aggregated chains shows stronger J-type character compared to single chains, a crucial result not demonstrated previously.

We have added a sentence and changed the following paragraph in the manuscript to clarify this point: *"Previous work on P3HT nanofibers demonstrated that interchain coupling, i.e. H-coupling, can be also altered by applying pressure or decreasing the temperature^{40, 41}.*

However, we conclude here that the predominant coupling type in mesoscopic aggregates can be discretely switched between H-type and J-type. This result further implies that both coupling mechanisms are present in the “dry” aggregates. The interchain H-type coupling is completely switched off by partially swelling the aggregates due to the increased distance between neighboring CP chains, leaving behind the intrachain J-type coupling. Surprisingly, we observe that the remaining PL of the aggregated chains shows stronger J-type character compared to single chains. Therefore, small changes in chain morphology can be responsible for large spectroscopic differences: controlling the morphology in the “dry” state becomes a crucial material parameter.”

5) The authors mentioned that the H-type emitters within the PPEb-2 aggregates may bleach with time, which leaves behind J-type of emitter. What is the reason for J-type to be more photo-resistance?

R4.5: The photostability of an emitter is mainly linked to its excited-state lifetime, because the excited state can undergo an electron transfer, which leaves behind a radical anion or cation state. These states are a precursor for further irreversible photodestruction. The J-type aggregate has a significantly shorter PL lifetime (<0.5 ns) as compared to the H-type aggregate (>3 ns), and should therefore be somewhat more photostable.

We have inserted a sentence to speculate on this point: *“H-type emitters within the PPEB-2 aggregates may bleach with time, leaving behind J-type emitters, which in turn may exhibit a higher degree of photostability compared to the H-type emitters. The longer excited-state lifetime of H-type emitters likely provides a higher probability to undergo irreversible photochemical reactions.”*

6) Several lines above the Methods, the author mentioned that the dark states are quenched by interchromophore (not really sure what it refers to) interactions in the H-aggregates, which is in analogy to the strong quenching of triplets in P3HT. I'm not sure whether in the PPE system, the quenching is also associate with triplet or not.

R4.6: We apologize for the lack of clarity. All we meant to reiterate here is that the dark state appears to vanish in the aggregate. We now write *“...are quenched by the interaction between chromophores in multichain aggregates”* to make this point clear. We agree that it is extremely hard to prove conclusively that the dark state quenched here really is the triplet, but we would prefer to leave the analogy of the observation to what is found in P3HT in the outlook of the work. Since this part of the manuscript is a somewhat speculative outlook, we think that it is formulated sufficiently cautiously.

REVIEWERS' COMMENTS:

Reviewer #4 (Remarks to the Author):

The authors have addressed most of the queries from the reviewers. For those without clear answers, justification has also been made. I'm satisfied with the revision and thus would like to recommend the acceptance of the manuscript.